# Point process models for sequence detection in high-dimensional neural spike trains

**Alex H. Williams**
Department of Statistics
Stanford University
Stanford, CA 94305
ahwillia@stanford.edu

**Anthony Degleris**
Department of Electrical Engineering
Stanford University
Stanford, CA 94305
degleris@stanford.edu

**Yixin Wang**
Department of Statistics
Columbia University
New York NY 10027
yixin.wang@columbia.edu

**Scott W. Linderman**
Department of Statistics
Stanford University
Stanford, CA 94305
scott.linderman@stanford.edu

## Abstract

Sparse sequences of neural spikes are posited to underlie aspects of working memory [1], motor production [2], and learning [3, 4]. Discovering these sequences in an unsupervised manner is a longstanding problem in statistical neuroscience [5–7]. Promising recent work [4, 8] utilized a convolutive nonnegative matrix factorization model [9] to tackle this challenge. However, this model requires spike times to be discretized, utilizes a sub-optimal least-squares criterion, and does not provide uncertainty estimates for model predictions or estimated parameters. We address each of these shortcomings by developing a point process model that characterizes fine-scale sequences at the level of individual spikes and represents sequence occurrences as a small number of marked events in continuous time. This ultra-sparse representation of sequence events opens new possibilities for spike train modeling. For example, we introduce learnable time warping parameters to model sequences of varying duration, which have been experimentally observed in neural circuits [10]. We demonstrate these advantages on experimental recordings from songbird higher vocal center and rodent hippocampus.

## 1   Introduction

Identifying interpretable patterns in multi-electrode recordings is a longstanding and increasingly pressing challenge in neuroscience. Depending on the brain area and behavioral task, the activity of large neural populations may be low-dimensional as quantified by principal components analysis (PCA) or other latent variable models [11–23]. However, many datasets do not conform to these modeling assumptions and instead exhibit high-dimensional behaviors [24]. *Neural sequences* are an important example of high-dimensional structure: if $N$ neurons fire sequentially with no overlap, the resulting dynamics are $N$-dimensional and cannot be efficiently summarized by PCA or other linear dimensionality reduction methods [4]. Such sequences underlie current theories of working memory [1, 25], motor production [2], and memory replay [10]. More generally, neural sequences are seen as flexible building blocks for learning and executing complex neural computations [26–28].

**Prior Work**   In practice, neural sequences are usually identified in a supervised manner by correlating neural firing with salient sensory cues or behavioral actions. For example, hippocampal place cells fire sequentially when rodents travel down a narrow hallway, and these sequences can be found

by averaging spikes times over multiple traversals of the hallway. After identifying this sequence on a behavioral timescale lasting seconds, a template matching procedure can be used to show that these sequences reoccur on compressed timescales during wake [29] and sleep [30]. In other cases, sequences can be identified relative to salient features of the local field potential (LFP; [31, 32]).

Developing unsupervised alternatives that directly extract sequences from multineuronal spike trains would broaden the scope of this research and potentially uncover new sequences that are not linked to behaviors or sensations [33]. Several works have shown preliminary progress in this direction Maboudi et al. [34] proposed fitting a hidden Markov model (HMM) and then identifying sequences from the state transition matrix. Grossberger et al. [35] and van der Meij & Voytek [36] apply clustering to features computed over a sliding window. Others use statistics such as time-lagged correlations to detect sequences and other spatiotemporal patterns in a bottom-up fashion [6, 7].

In this paper, we develop a Bayesian point process modeling generalization of *convolutive nonnegative matrix factorization* (convNMF; [9]), which was recently used by Peter et al. [8] and Mackevicius et al. [4] to model neural sequences. Briefly, the convNMF model discretizes each neuron's spike train into a vector of $B$ time bins, $\mathbf{x}_n \in \mathbb{R}_+^B$ for neuron $n$, and approximates this collection of vectors as a sum of convolutions, $\mathbf{x}_n \approx \sum_{r=1}^R \mathbf{w}_{n,r} * \mathbf{h}_r$. The model parameters ($\mathbf{w}_{n,r} \in \mathbb{R}_+^L$ and $\mathbf{h}_r \in \mathbb{R}_+^B$) are optimized with respect to a least-squares criterion. Each component of the model, indexed by $r \in \{1, \ldots, R\}$, consists of a *neural factor*, $\mathbf{W}_r \in \mathbb{R}_+^{N \times L}$ and a *temporal factor*, $\mathbf{h}_r$. The neural factor encodes a spatiotemporal pattern of neural activity over $L$ time bins (which is hoped to be sequence), while the temporal factor indicates the times at which this pattern of neural activity occurs. A total of $R$ motifs or sequence types, each corresponding to a different component, are extracted by this model.

There are several compelling features of this approach. Similar to classical nonnegative matrix factorization [37], and in contrast to clustering methods, convNMF captures sequences with overlapping groups of neurons by an intuitive "parts-based" representation. Indeed, convNMF uncovered overlapping sequences in experimental data from songbird higher vocal center (HVC) [4]. Further, if the same sequence is repeated with different peak firing rates, convNMF can capture this by varying the magnitude of the entries in $\mathbf{h}_r$, unlike many clustering methods. Finally, convNMF efficiently pools statistical power across large populations of neurons to identify sequences even when the correlations between temporally adjacent neurons are noisy—other methods, such as HMMs and bottom-up agglomerative clustering, require reliable pairwise correlations to string together a full sequence.

**Our Contributions**  We propose a point process model for neural sequences (PP-Seq) which extends and generalizes convNMF to continuous time and uses a fully probabilistic Bayesian framework. This enables us to better quantify uncertainty in key parameters—e.g. the overall number of sequences the times at which they occur—and also characterize the data at finer timescales—e.g. whether individual spikes were evoked by a sequence. Most importantly, by achieving an extremely sparse representation of sequence event times, the PP-Seq model enables a variety of model extensions that are not easily incorported into convNMF or other common methods. We explore one such extension that introduces time warping factors to model sequences of varying duration, as is often observed in neural data [10].

Though we focus on applications in neuroscience, our approach could be adapted to other temporal point processes, which are a natural framework to describe data that are collected at irregular intervals (e.g. social media posts, consumer behaviors, and medical records) [38, 39]. We draw a novel connection between Neyman-Scott processes [40], which encompass PP-Seq and other temporal point process models as special cases, and mixture of finite mixture models [41]. Exploiting this insight, we develop innovative Markov chain Monte Carlo (MCMC) methods for PP-Seq.

## 2 Model

### 2.1 Point Process Models and Neyman-Scott Processes

Point processes are probabilistic models that generate discrete sets $\{x_1, x_2, \ldots\} \triangleq \{x_s\}_{s=1}^S$ over some continuous space $\mathcal{X}$. Each member of the set $x_s \in \mathcal{X}$ is called an "event." A Poisson process [42] is a point process which satisfies three properties. First, the number of events falling within any region $V \subset \mathcal{X}$ is Poisson distributed. Second, there exists a function $\lambda(x) : \mathcal{X} \mapsto \mathbb{R}_+$, called the *intensity function*, for which $\int_V \lambda(x) \, \mathrm{d}x$ equals the expected number of events in $V$. Third,

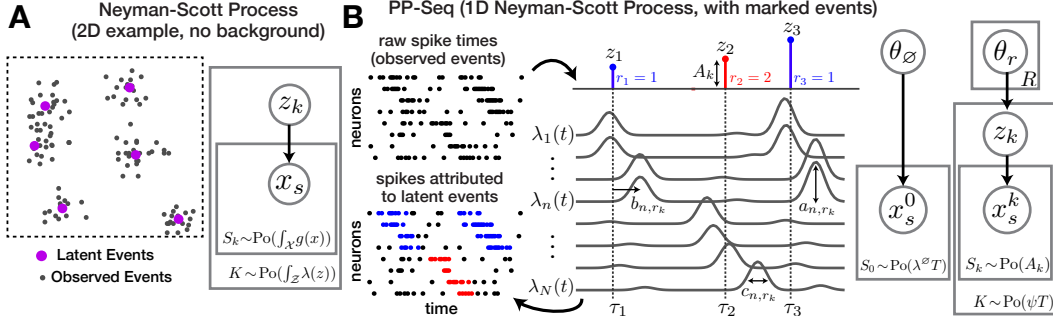

Figure 1: *(A)* Example of a Neyman-Scott process over a 2D region. Latent events (purple dots) are first sampled from a homogeneous Poisson process. Each latent event then spawns a number of nearby observed events (gray dots), according to an inhomogeneous Poisson process. *(B)* A spike train can be modeled as a Neyman-Scott process with *marked* events over a 1D interval representing time. Latent events ($z_k$; sequences) evoke observed events ($x_s$; spikes) ordered in a sequence. An example with $K = 3$ latent events evoking $R = 2$ different sequence types (blue & red) is shown.

the number of events in non-overlapping regions of $\mathcal{X}$ are independent. A Poisson process is said to be *homogeneous* when $\lambda(x) = c$ for some constant $c$. Finally, a *marked* point process extends the usual definition of a point process to incorporate additional information into each event. A marked point process on $\mathcal{X}$ generates random tuples $x_s = (\tilde{x}_s, m_s)$, where $\tilde{x}_s \in \mathcal{X}$ represents the random location and $m_s \in \mathcal{M}$ is the additional "mark" specifying metadata associated with event $s$. See Supplement A for further background details and references.

*Neyman-Scott Processes* [40] use Poisson processes as building blocks to model clustered data. To simulate a Neyman-Scott process, first sample a set of *latent events* $\{z_k\}_{k=1}^K \subset \mathcal{Z}$ from an initial Poisson process with intensity function $\lambda(z) : \mathcal{Z} \mapsto \mathbb{R}_+$. Thus, the number of latent events is a Poisson-distributed random variable, $K \sim \text{Poisson}(\int_{\mathcal{Z}} \lambda(z) \mathrm{d}z)$. Given the latent events, the *observed events* $\{x_s\}_{s=1}^S$ are drawn from a second Poisson process with conditional intensity function $\lambda(x) = \lambda^{\varnothing} + \sum_{k=1}^K g(x; z_k)$. The nonnegative functions $g(x; z_k)$ can be thought of as impulse responses—each latent event adds to the rate of observed events, and stochastically generates some number of observed offspring events. Finally, the scalar parameter $\lambda^{\varnothing} > 0$ specifies a "background" rate; thus, if $K = 0$, the observations follow a homogeneous Poisson process with intensity $\lambda^{\varnothing}$.

A simple example of a Neyman-Scott Process in $\mathbb{R}^2$ is shown in fig. 1A. Latent events (purple dots) are first drawn from a homogenous Poisson process and specify the cluster centroids. Each latent event induces an isotropic Gaussian impulse response. Observed events (gray dots) are then sampled from a second Poisson process, whose intensity function is found by summing all impulse responses.

Importantly, we *do not* observe the number of latent events nor their locations. As described below, we use probabilistic inference to characterize this unobserved structure. A key idea is to attribute each observed event to a latent cause (either one of the latent events or the background process); these attributions are valid due to the additive nature of the latent intensity functions and the *superposition principle* of Poisson processes (see Supplement A). From this perspective, inferring the set of latent events is similar to inferring the number and location of clusters in a nonparametric mixture model.

## 2.2 Point Process Model of Neural Sequences (PP-Seq)

We model neural sequences as a Neyman-Scott process with *marked* events in a model we call PP-Seq. Consider a dataset with $N$ neurons, emitting a total of $S$ spikes over a time interval $[0, T]$. This can be encoded as a set of $S$ marked spike times—the observed events are tuples $x_s = (t_s, n_s)$ specifying the time, $t_s \in [0, T]$, and neuron, $n_s \in \{1, \ldots, N\}$, of each spike. Sequences correspond to latent events, which are also tuples $z_k = (\tau_k, r_k, A_k)$ specifying the time, $\tau_k \in [0, T]$, type, $r_k \in \{1, \ldots, R\}$, and amplitude, $A_k > 0$ of the sequence. The hyperparameter $R$ specifies the number of recurring sequence types, which is analogous to the number of components in convNMF.

To draw samples from the PP-Seq model, we first sample sequences (i.e., latent events) from a Poisson process with intensity $\lambda(z) \triangleq \lambda(\tau, r, A) = \psi \, \pi_r \, \text{Ga}(A; \alpha, \beta)$. Here, $\psi > 0$ sets the rate at

which sequences occur within the data, $\pi \in \Delta_R$ sets the probability of the $R$ sequence types, and $\alpha, \beta$ parameterize a gamma density which models the sequence amplitude, $A$. Note that the number of sequence events is a Poisson-distributed random variable, $K \sim \text{Poisson}(\psi T)$, where the rate parameter $\psi T$ is found by integrating $\lambda(z)$ over all sequence types, amplitudes, and times.

Conditioned on the set of $K$ sequence events, the firing rate of neuron $n$ is given by a sum of nonnegative impulse responses:

$$\lambda_n(t) = \lambda_n^{\varnothing} + \sum_{k=1}^{K} g_n(t; z_k). \tag{1}$$

We assume these impulse responses vary across neurons and follow a Gaussian form:

$$g_n(t; z_k) = A_k \cdot a_{nr_k} \cdot \mathcal{N}(t \mid \tau_k + b_{nr_k}, c_{nr_k}), \tag{2}$$

where $\mathcal{N}(t \mid \mu, \sigma^2)$ denotes a Gaussian density. The parameters $a_r = (a_{1r}, \ldots, a_{Nr}) \in \Delta_N$, $b_{nr} \in \mathbb{R}$, and $c_{nr} \in \mathbb{R}_+$ correspond to the weight, latency, and width, respectively, of neurons' firing rates in sequences of type $r$. Since the firing rate is a sum of non-negative impulse responses, the superposition principle of Poisson processes (see Supplement A) implies that we can view the data as a union of "background" spikes and "induced" spikes from each sequence, justifying the connection to clustering. The expected number of spikes induced by sequence $k$ is:

$$\sum_{n=1}^{N} \int_0^T g_n(t; z_k)\mathrm{d}t \approx \sum_{n=1}^{N} \int_{-\infty}^{\infty} g_n(t; z_k)\mathrm{d}t = A_k, \tag{3}$$

and thus we may view $A_k$ as the amplitude of sequence event $k$.

Figure 1B schematizes a simple case containing $K = 3$ sequence events and $R = 2$ sequence types. A complete description of the model's generative process is provided in Supplement B, but it can be summarized by the graphical model in fig. 1B, where we have global parameters $\Theta = (\theta_{\varnothing}, \{\theta_r\}_{r=1}^R)$ with $\theta_r = (a_r, \{b_{nr}\}_{n=1}^N, \{c_{nr}\}_{n=1}^N)$ for each sequence type, and $\theta_{\varnothing} = \{\lambda_n^{\varnothing}\}_{n=1}^N$ for the background process. We place weak priors on each parameter: the neural response weights $\{a_{nr}\}_{n=1}^N$ follow a Dirichlet prior for each sequence type, and $(b_{nr}, c_{nr})$ follows a normal-inverse-gamma prior for every neuron and sequence type. The background rate, $\lambda_n^{\varnothing}$, follows a gamma prior. We set the sequence event rate, $\psi$, to be a fixed hyperparameter, though this assumption could be relaxed.

**Time-warped sequences**   PP-Seq can be extended to model more diverse sequence patterns by using higher-dimensional marks on the latent sequences. For example, we can model variability in sequence duration by introducing a *time warping factor*, $\omega_k > 0$, to each sequence event and changing eq. (2) to,

$$g_n(t; z_k) = A_k \cdot a_{n,r_k} \cdot \mathcal{N}(t \mid \tau_k + \omega_k b_{n,r_k}, \omega_k^2 c_{n,r_k}). \tag{4}$$

This has the effect of linearly compressing or stretching each sequence in time (when $\omega_k < 1$ or $\omega_k > 1$, respectively). Such time warping is commonly observed in neural data [43, 44], and indeed, hippocampal sequences unfold ~15-20 times faster during replay than during lived experiences [10]. We characterize this model in Supplement E and demonstrate its utility below.

In principle, it is equally possible to incorporate time warping into discrete time convNMF. However, since convNMF involves a dense temporal factor matrix $\mathbf{H} \in \mathbb{R}_+^{R \times B}$, the most straightforward extension would be to introduce a time warping factor for each component $r \in \{1, \ldots, R\}$ and each time bin $b \in \{1, \ldots, B\}$. This results in $O(RB)$ new trainable parameters, which poses non-trivial challenges in terms of computational efficiency, overfitting, and human interpretability. In contrast, PP-Seq represents sequence events as a set of $K$ latent events in continuous time. This ultra-sparse representation of sequence events (since $K \ll RB$) naturally lends itself to modeling additional sequence features since this introduces only $O(K)$ new parameters.

## 3   Collapsed Gibbs Sampling for Neyman-Scott Processes

Developing efficient algorithms for parameter inference in Neyman-Scott process models is an area of active research [45–48]. To address this challenge, we developed a collapsed Gibbs sampling routine for Neyman-Scott processes, which encompasses the PP-Seq model as a special case. The method

resembles "Algorithm 3" of Neal [49]—a well-known approach for sampling from a Dirichlet process mixture model—and the collapsed Gibbs sampling algorithm for "mixture of finite mixtures" models developed by Miller et al. [41]. The idea is to partition observed spikes into background spikes and spikes induced by latent sequences, integrating over the sequence times, types, and amplitudes. Starting from an initial partition, the sampler iterates over individual spikes and probabilistically re-assigns them to *(a)* the background, *(b)* one of the remaining sequences, or *(c)* to a new sequence. The number of sequences in the partition, $K^*$, changes as spikes are removed and re-assigned; thus, the algorithm is able to explore the full trans-dimensional space of partitions.

The re-assignment probabilities are determined by the prior distribution of partitions under the Neyman-Scott process and by the likelihood of the induced spikes assigned to each sequence. We state the conditional probabilities below and provide a full derivation in Supplement D. Let $K^*$ denote the number of sequences in the current partition after spike $x_s$ has been removed from its current assignment. (Note that the number of latent sequences $K$ may exceed $K^*$ if some sequences produce zero spikes.) Likewise, let $u_s$ denote the sequence assignment of the $s$-th spike, where $u_s = 0$ indicates assignment to the background process and $u_s \in \{1, \ldots, K^*\}$ indicates assignment to one of the latent sequence events. Finally, let $X_k = \{x_{s'} : u_{s'} = k, s' \neq s\}$ denote the spikes in the $k$-th cluster, excluding $x_s$, and let $S_k = |X_k|$ denote its size. The conditional probability of the partition under the possible assignments of spike $x_s$ are,

$$p(u_s = 0 \mid x_s, \{X_k\}_{k=1}^{K^*}, \Theta) \propto (1 + \beta) \lambda_{n_s}^{\varnothing} \tag{5}$$

$$p(u_s = k \mid x_s, \{X_k\}_{k=1}^{K^*}, \Theta) \propto (\alpha + S_k) \left[ \sum_{r_k=1}^{R} p(r_k \mid X_k) \, a_{n_s r_k} \, p(t_s \mid X_k, r_k, n_s) \right] \tag{6}$$

$$p(u_s = K^* + 1 \mid x_s, \{X_k\}_{k=1}^{K^*}, \Theta) \propto \alpha \left( \frac{\beta}{1+\beta} \right)^{\alpha} \psi \sum_{r=1}^{R} \pi_r \, a_{n_s r} \tag{7}$$

The sampling algorithm iterates over all spikes $x_s \in \{x_1, \ldots, x_S\}$ and updates their assignments holding the other spikes' assignments fixed. The probability of assigning spike $x_s$ to an existing cluster marginalizes the time, type, and amplitude of the sequence, resulting in a *collapsed* Gibbs sampler [49, 50]. The exact form of the posterior probability $p(r_k \mid X_k)$ and the parameters of the posterior predictive $p(t_s \mid X_k, r_k, n_s)$ in eq. (6) are given in Supplement C.

After attributing each spike to a latent cause, it is straightforward to draw samples over the remaining model parameters—the latent sequences $\{z_k\}_{k=1}^{K^*}$ and global parameters $\Theta$. Given the spikes and assignments $\{x_s, u_s\}_{s=1}^{S}$, we sample the sequences (i.e. their time, amplitude, types, etc.) from the closed-form conditional $p(z_k \mid \{x_s : u_s = k\}, \Theta)$. Given the sequences and spikes, we sample the conditional distribution on global parameters $p(\Theta \mid \{z_k\}_{k=1}^{K^*}, \{x_s, u_s\}_{s=1}^{S})$. Under conjugate formulations, these updates are straightforward. With these steps, the Markov chain targets the posterior distribution on model parameters and partitions. Complete derivations are in Supplement D.

**Improving MCMC mixing times**  The intensity of sequence amplitudes $A_k$ is proportional to the gamma density $\mathrm{Ga}(A_k; \alpha, \beta)$, and these hyperparameters affect the mixing time of the Gibbs sampler. Intuitively, if there is little probability of low-amplitude sequences, the sampler is unlikely to create new sequences and is therefore slow to explore different partitions of spikes.[1] If, on the other hand, the variance of $\mathrm{Ga}(\alpha, \beta)$ is large relative to the mean, then the probability of forming new clusters is non-negligible and the sampler tends to mix more effectively. Unfortunately, this latter regime is also probably of lesser scientific interest, since neural sequences are typically large in amplitude—they can involve many thousands of cells, each potentially contributing a small number of spikes [2, 28].

To address this issue, we propose an annealing procedure to initialize the Markov chain. We fix the mean of $A_k$ and adjust $\alpha$ and $\beta$ to slowly lower variance of amplitude distribution. Initially, the sampler produces many small clusters of spikes, and as we lower the variance of $\mathrm{Ga}(\alpha, \beta)$ to its target value, the Markov chain typically combines these clusters into larger sequences. We further improve performance by interspersing "split-merge" Metropolis-Hastings updates [51, 52] between Gibbs sweeps (see Supplement D.6). Finally, though we have not found it necessary, one could use convNMF to initialize the MCMC algorithm.

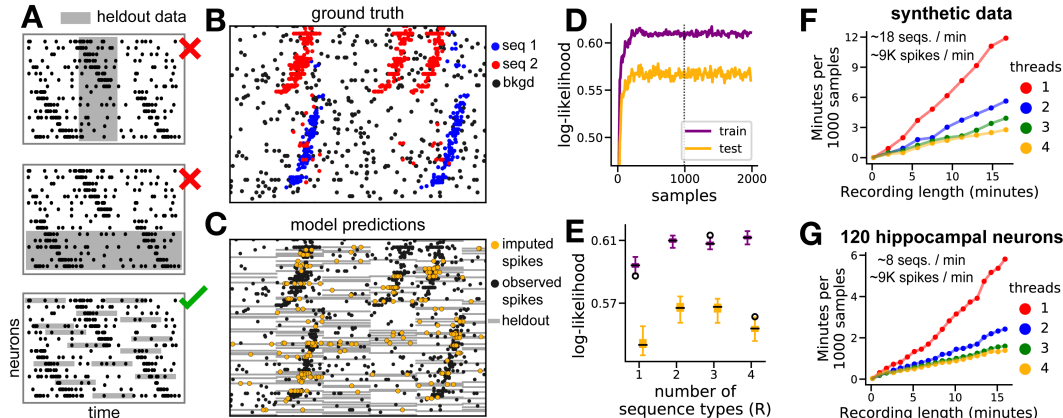

Figure 2: *(A)* Schematic of train/test partitions. We propose a speckled holdout pattern (bottom). *(B)* A subset of a synthetic spike train containing two sequences types. *(C)* Same data, but with grey regions showing the censored test set and yellow dots denoting imputed spikes. *(D)* Log-likelihood over Gibbs samples; positive values denote excess nats per unit time relative to a homogeneous Poisson process baseline. *(E)* Box plots showing range of log-likelihoods on the train and test sets for different choices of $R$; cross-validation favors $R = 2$, in agreement with the ground truth shown in panel B. *(F-G)* Performance benefits of parallel MCMC on synthetic and experimental neural data.

**Parallel MCMC** Resampling the sequence assignments is the primary computational bottleneck for the Gibbs sampler. One pass over the data requires $O(SKR)$ operations, which quickly becomes costly when the operations are serially executed. While this computational cost is manageable for many datasets, we can improve performance substantially by parallelizing the computation [53]. Given $P$ processors and a spike train lasting $T$ seconds, we divide the dataset into intervals lasting $T/P$ seconds, and allocate one interval per processor. The current global parameters, $\Theta$, are first broadcast to all processors. In parallel, the processors update the sequence assignments for their assigned spikes, and then send back sufficient statistics describing each sequence. After these sufficient statistics are collected on a single processor, the global parameters are re-sampled and then broadcast back to the processors to initiate another iteration. This algorithm introduces some error since clusters are not shared across processors. In essence, this introduces erroneous edge effects if a sequence of spikes is split across two processors. However, these errors are negligible when the sequence length is much less than $T/P$, which we expect is the practical regime of interest.

## 4 Experiments

### 4.1 Cross-Validation and Demonstration of Computational Efficiency

We evaluate model performance by computing the log-likelihood assigned to held-out data. Partitioning the data into training and testing sets must be done somewhat carefully—we cannot withhold time intervals completely (as in fig. 2A, *top*) or else the model will not accurately predict latent sequences occurring in these intervals; likewise, we cannot withhold individual neurons completely (as in fig. 2A, *middle*) or else the model will not accurately predict the response parameters of those held out cells. Thus, we adopt a "speckled" holdout strategy [54] as diagrammed at the bottom of fig. 2A. We treat held-out spikes as missing data and sample them as part of the MCMC algorithm. (Their conditional distribution is given by the PP-Seq generative model.) This approach involving a speckled holdout pattern and multiple imputation of missing data may be viewed as a continuous time extension of the methods proposed by Mackevicius et al. [27] for convNMF.

Panels B-E in fig. 2 show the results of this cross-validation scheme on a synthetic dataset with $R = 2$ sequence types. The predictions of the model in held-out test regions closely match the ground truth—missing spikes are reliably imputed when they are part of a sequence (fig. 2C). Further, the likelihood of the train and test sets improves over the course of MCMC sampling (fig. 2D), and can be used as a metric for model comparison—in agreement with the ground truth, test performance plateaus for models containing greater than $R = 2$ sequence types (fig. 2E).

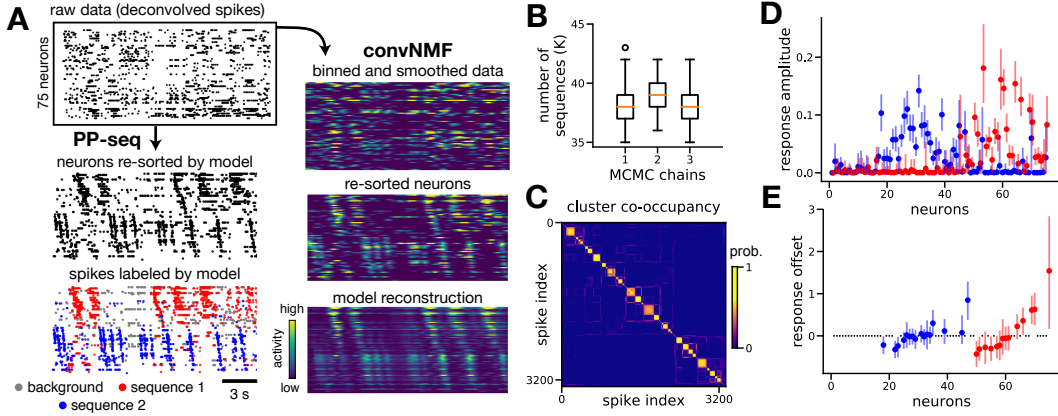

Figure 3: Zebra Finch HVC data. *(A)* Raw spike train (top) and sequences revealed by PP-Seq (left) and convNMF (right). *(B)* Box plots summarizing samples from the posterior on number of sequences, $K$, derived from three independent MCMC chains. *(C)* Co-occupancy matrix summarizing probabilities of spike pairs belonging to the same sequence. *(D)* Credible intervals for evoked amplitudes for sequence type 1 (red) and 2 (blue). *(E)* Credible intervals for response offsets (same order and coloring as *D*). Estimates are suppressed for small-amplitude responses (gray dots).

Finally, to be of practical utility, the algorithm needs to run in a reasonable amount of time. Figure 2G shows that our Julia [55] implementation can fit a recording of 120 hippocamapal neurons with hundreds of thousands of spikes in a matter of minutes, on a 2017 MacBook Pro (3.1 GHz Intel Core i7, 4 cores, 16 GB RAM). Run-time grows linearly with the number of spikes, as expected, but even with a single thread it only takes six minutes to perform 1000 Gibbs sweeps on a 15-minute recording with $\sim 1.3 \times 10^5$ spikes. With parallel MCMC, this laptop performs the same number of sweeps in under two minutes. Our open-source implementation is available at:

$$\texttt{https://github.com/lindermanlab/PPSeq.jl.}$$

## 4.2 Zebra Finch Higher Vocal Center (HVC)

We first applied PP-Seq to a recording of HVC premotor neurons in a zebra finch,[2] which generate sequences that are time-locked to syllables in the bird's courtship song. Figure 3A qualitatively compares the performance of convNMF and PP-Seq. The raw data (top panel) shows no visible spike patterns; however, clear sequences are revealed by sorting the neurons lexographically by preferred sequence type and the temporal offset parameter inferred by PP-Seq. While both models extract similar sequences, PP-Seq provides a finer scale annotation of the final result, providing, for example, attributions at the level of individual spikes to sequences (bottom left of fig. 3A).

Further, PP-Seq can quantify uncertainty in key parameters by considering the full sequence of MCMC samples. Figure 3B summarizes uncertainty in the total number of sequence events, i.e. $K$, over three independent MCMC chains with different random seeds—all chains converge to similar estimates; the uncertainty is largely due to the rapid sequences (in blue) shown in panel A. Figure 3C displays a symmetric matrix where element $(i, j)$ corresponds to the probability that spike $i$ and spike $j$ are attributed to same sequence. Finally, fig. 3D-E shows the amplitude and offset for each neuron's sequence-evoked response with 95% posterior credible intervals. These results naturally fall out of the probabilistic construction of the PP-Seq model, but have no obvious analogue in convNMF.

## 4.3 Time Warping Extension and Robustness to Noise

Songbird HVC is a specialized circuit that generates unusually clean and easy-to-detect sequences. To compare the robustness of PP-Seq and convNMF under more challenging circumstances, we created a simple synthetic dataset with $R = 1$ sequence type and $N = 100$ neurons. We varied four parameters to manipulate the difficulty of sequence extraction: the rate of background spikes, $\lambda_n^\varnothing$ ("additive

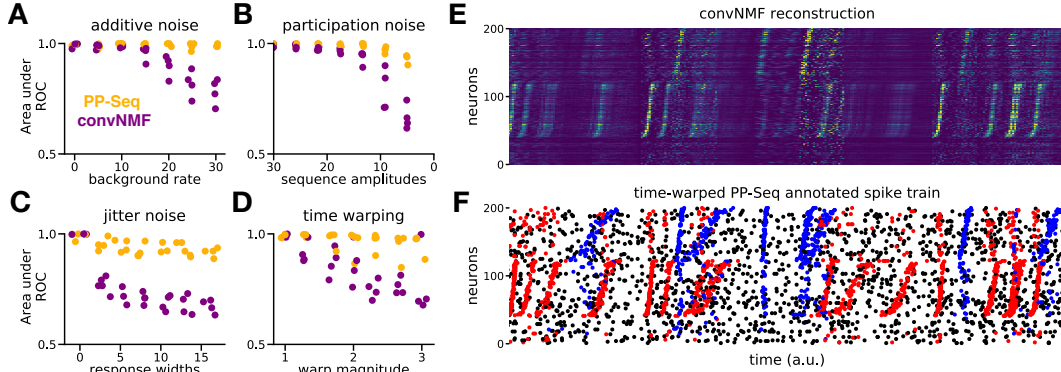

Figure 4: PP-Seq is more robust to various forms of noise than convNMF. *(A-D)* Comparison of PP-Seq and convNMF to detect sequence times in synthetic data with varying amounts of noise. Panel D shows the performance of the time-warped variant of PP-Seq (see eq. (4)). *(E)* convNMF reconstruction of a spike train containing sequences with 9-fold time warping. *(F)* Performance of time-warped PP-Seq (see eq. (4)) on the same data as panel E.

noise," fig. 4A), the expected value of sequence amplitudes, $A_k$ ("participation noise", fig. 4B), the expected variance of the Gaussian impulse responses, $c_{nr}$ ("jitter noise", fig. 4C), and, finally, the maximal time warping coefficient (see eq. (4); fig. 4D). All simulated datasets involved sequences with low spike counts ($\mathbb{E}[A_k] < 100$ spikes). In this regime, the Poisson likelihood criterion used by PP-Seq is better matched to the statistics of the spike train. Since convNMF optimizes an alternative loss function (squared error instead of Poisson likelihood) we compared the models by their ability to extract the ground truth sequence event times. Using area under reciever operating characteriztic (ROC) curves as a performance metric (see Supplement F.1), we see favorable results for PP-Seq as noise levels are increased.

We demonstrate the abilities of time-warped PP-Seq further in fig. 4E-F. Here, we show a synthetic dataset containing sequences with 9-fold variability in their duration, which is similar to levels observed in some experimental systems [10]. While convNMF fails to reconstruct many of these warped sequences, the PP-Seq model identifies sequences that are closely matched to ground truth.

## 4.4 Rodent Hippocampal Sequences

Finally, we tested PP-Seq and its time warping variant on a hippocampal recording in a rat making repeated runs down a linear track.[3] This dataset is larger ($T \approx 16$ minutes, $S = 137{,}482$) and contains less stereotyped sequences than the songbird data. From prior work [56], we expect to see two sequences with overlapping populations of neurons, corresponding to the two running directions on the track. PP-Seq reveals these expected sequences in an unsupervised manner—i.e. without reference to the rat's position—as shown in fig. 5A-C.

We performed a large cross-validation sweep over 2,000 random hyperparameter settings for this dataset (see Supplement F.2). This confirmed that models with $R = 2$ sequence performed well in terms of heldout performance (fig. 5D). Interestingly, despite variability in running speeds, this same analysis did not show a consistent benefit to including larger time warping factors into the model (fig. 5E). Higher performing models were characterized by larger sequence amplitudes, i.e. larger values of $\mathbb{E}[A_k] = \alpha/\beta$, and smaller background rates, i.e. smaller values of $\lambda^\varnothing$. Other parameters had less pronounced effects on performance. Overall, these results demonstrate that PP-Seq can be fruitfully applied to large-scale and "messy" neural datasets, that hyperparameters can be tuned by cross-validation, and that the unsupervised learning of neural sequences conforms to existing scientific understanding gained via supervised methods.

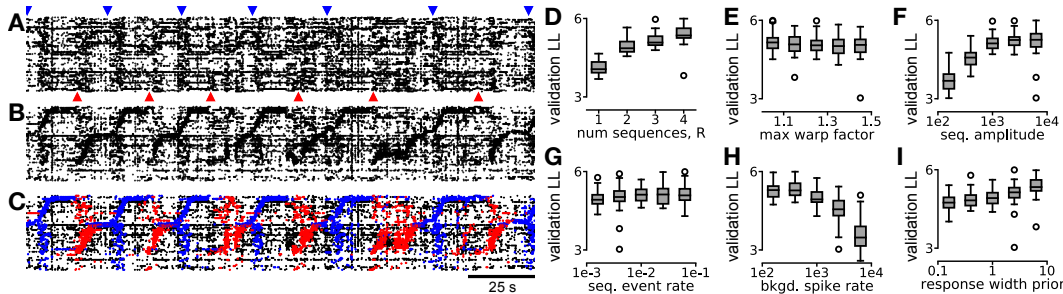

Figure 5: Automatic detection of place coding sequences in rat hippocampus. *(A)* Raw spike train ($\sim$20% of the full dataset is shown). Blue and red arrowheads indicate when the mouse reaches the end of the track and reverses direction. *(B)* Neurons re-sorted by PP-Seq (with $R = 2$). *(C)* Sequences annotated by PP-Seq. *(D-I)* Validation log-likelihoods as a function of hyperparameter value. Each boxplot summarizes the 50 highest scoring models, randomly sampling the other hyperparameters.

## 5  Conclusion

We proposed a point process model (PP-Seq) inspired by convolutive NMF [4, 8, 9] to identify neural sequences. Both models approximate neural activity as the sum of nonnegative features, drawn from a fixed number of spatiotemporal motif types. Unlike convNMF, PP-Seq restricts each motif to be a true sequence—the impulse responses are Gaussian and hence unimodal. Further, PP-Seq is formulated in a probabilistic framework that better quantifies uncertainty (see fig. 3) and handles low firing rate regimes (see fig. 4). Finally, PP-Seq produces an extremely sparse representation of sequence events in continuous time, opening the door to a variety of model extensions including the introduction of time warping (see fig. 4F), as well as other possibilities like truncated sequences and "clusterless" observations [57], which could be explored in future work.

Despite these benefits, fitting PP-Seq involves a tackling a challenging trans-dimensional inference problem inherent to Neyman-Scott point processes. We took several important steps towards overcoming this challenge by connecting these models to a more general class of Bayesian mixture models [41], developing and parallelizing a collapsed Gibbs sampler, and devising an annealed sampling approach to promote fast mixing. These innovations are sufficient to fit PP-Seq on datasets containing hundreds of thousands of spikes in just a few minutes on a modern laptop.

**Acknowledgements**

A.H.W. received funding support from the National Institutes of Health BRAIN initiative (1F32MH122998-01), and the Wu Tsai Stanford Neurosciences Institute Interdisciplinary Scholar Program. S.W.L. was supported by grants from the Simons Collaboration on the Global Brain (SCGB 697092) and the NIH BRAIN Initiative (U19NS113201 and R01NS113119). We thank the Stanford Research Computing Center for providing computational resources and support that contributed to these research results.

**Broader Impact**

Understanding neural computations in biological systems and ultimately the human brain is a grand and long-term challenge with broad implications for human health and society. The field of neuroscience is still taking early and incremental steps towards this goal. Our work develops a general-purpose, unsupervised method for identifying an important structure—neural sequences—which have been observed in a variety of experimental datasets and have been studied extensively by theorists. This work will serve to advance this growing understanding by providing new analytical tools for neuroscientists. We foresee no immediate impacts, positive or negative, concerning the general public.

## Footnotes

[1]This problem is common to other nonparametric Bayesian mixture models as well [41, e.g.].

[2]These data are available at `http://github.com/FeeLab/seqNMF`; originally published in [4].

[3] These data are available at `http://crcns.org/data-sets/hc/hc-11`; originally published in [56].

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
