[Supplementary Material]

# Supplementary Material: Point process models for sequence detection in high-dimensional spike trains

## A    Background

This section describes some basic properties and characteristics of Poisson processes that are relevant to understand the PP-Seq model. See Kingman [1] for a targeted introduction to Poisson processes and Daley et al. [2] for a broader introduction to point processes.

**Definition of a Poisson Process** — A point process model satisfying the following three properties is called a *Poisson process*.

  (i) The number of events falling within any region of the sample space $\mathcal{X}$ is a random variable following a Poisson distribution. For a region $R \subset \mathcal{X}$ we denote the number of events occuring in $R$ as $N(R)$.

  (ii) There exists a function $\lambda(x) : \mathcal{X} \mapsto \mathbb{R}_+$, called the *intensity function*, which satisfies $N(R) \sim \text{Poisson}(\int_R \lambda(x)dx)$ for every region $R \subset \mathcal{X}$.

  (iii) For any two non-overlapping regions $R$ and $R'$, $N(R)$ and $N(R')$ are independent random variables.

We write $\{x_i\}_{i=1}^n \sim PP(\lambda)$ to denote that a set of events $\{x_i\}_{i=1}^n = \{x_1, \ldots, x_n\}$ is distributed according to a Poisson process with intensity function $\lambda$.

**Marked Poisson Processes** — A *marked* Poisson process extends the above definition to sets of events marked with additional information. For example, we write $\{(x_i, m_i)\}_{i=1}^n \sim PP(\lambda(x, m))$ to denote a marked Poisson process where $m_i \in \mathcal{M}$ is the "mark" associated with each event. Note that the domain of the intensity function is extended to $\mathcal{X} \times \mathcal{M}$. To pick a concrete example of interest, the events could represent spike times, in which case $\mathcal{X}$ would be an interval of the real line, e.g. $[0, T]$, representing the recorded period. Further, $\mathcal{M}$ would be a discrete set of neuron labels, e.g. $\{1, \ldots, N\}$ for a recording of $N$ cells.

**Likelihood function** — Given a set of events $\{x_i\}_{i=1}^n$ in $\mathcal{X}$ the likelihood of this set, with respect to Poisson process $PP(\lambda)$ is:

$$p(\{x_i\}_{i=1}^n \mid \lambda) = \prod_{i=1}^n \lambda(x_i) \cdot \exp \left\{ - \int_{\mathcal{X}} \lambda(x)dx \right\} \tag{8}$$

**Sampling** — Given an intensity function $\lambda(x)$ and a sample space $\mathcal{X}$, we can simulate data from a Poisson process by first sampling the number of events:

$$N \sim \text{Poisson} \left( \int_{\mathcal{X}} \lambda(x)dx \right) \tag{9}$$

and then sampling the location of each event according to the normalized intensity function. That is, for $i = \{1, \ldots, N\}$, we independently sample the events according to:

$$x_i \sim F \quad \text{where the p.d.f. of } F \text{ is } \frac{\lambda(x)}{\int_{\mathcal{X}} \lambda(x)dx} \tag{10}$$

**Superposition Principle** — The union of $K$ independent Poisson processes with nonnegative intensity functions $\lambda_1(x), \ldots, \lambda_K(x)$ is a Poisson process with intensity function $\Lambda(x) = \sum_{k=1}^K \lambda_k(x)$.

As a consequence, we can sample from a complicated Poisson process by first breaking it down into a series of simpler Poisson processes, drawing independent samples from these simpler processes, and taking the union of all events to achieve a sample from the original Poisson process. We exploit this property in section B.1 to draw samples from the PP-Seq model.

## B    The PP-Seq Model

### B.1    Generative Process

Recall the generative model described in the main text. Let $z_k = (\tau_k, r_k, A_k)$ denote the $k$-th latent sequence, which consists of a time $\tau_k$, type $r_k$, and amplitude $A_k$. Extensions of the model introduce further properties like time warping

factors $\omega_k$ for each latent sequence. The latent events (i.e. instantiations of a sequence) are sampled from a Poisson process,

$$\{(\tau_k, r_k, A_k)\}_{k=1}^K \sim \mathrm{PP}\big(\lambda(\tau, r, A)\big) \tag{11}$$

$$\lambda(\tau, r, A) = \psi\, \pi_r\, \mathrm{Ga}(A; \alpha, \beta). \tag{12}$$

Note that this is a *homogeneous* Poisson process since the intensity function does not depend on time. Conditioned on the latent events, the observed spikes are then sampled from a second, inhomogeneous, Poisson process,

$$\{(t_s, n_s)\}_{s=1}^S \sim \mathrm{PP}\big(\lambda(t, n \mid \{(\tau_k, r_k, A_k)\}_{k=1}^K)\big) \tag{13}$$

$$\lambda(t, n \mid \{(\tau_k, r_k, A_k)\}_{k=1}^K) = \lambda^\varnothing(t, n) + \sum_{k=1}^K g(t, n \mid \tau_k, r_k, A_k) \tag{14}$$

$$g(t, n \mid \tau_k, r_k, A_k) = A_k \cdot a_{nr_k} \cdot \mathcal{N}(t \mid \tau_k + b_{nr_k}, c_{nr_k}). \tag{15}$$

In the main text we abbreviate $\lambda_n(t) \triangleq \lambda(t, n \mid \{(\tau_k, r_k, A_k)\}_{k=1}^K)$, $\lambda^\varnothing(t, n) \triangleq \lambda_n^\varnothing$, and $g_n(t \mid z_k) \triangleq g(t, n \mid \tau_k, r_k, A_k)$.

Observe that the firing rates are sums of non-negative intensity functions. We can use the superposition principle of Poisson processes (see section A) to write the observed spikes as the union of spikes generated by the background and by each of the latent events.

$$\{(t_s^{(0)}, n_s^{(0)})\}_{s=1}^{S_0} \sim \mathrm{PP}\big(\lambda^\varnothing(t, n)\big) \tag{16}$$

$$\{(t_s^{(k)}, n_s^{(k)})\}_{s=1}^{S_k} \sim \mathrm{PP}\big(A_k \cdot a_{nr_k} \cdot \mathcal{N}(t \mid \tau_k + b_{nr_k}, c_{nr_k})\big) \qquad \text{for } k = 1, \ldots, K \tag{17}$$

$$\{(t_s, n_s)\}_{s=1}^S = \bigcup_{k=0}^K \{(t_s^{(k)}, n_s^{(k)})\}_{s=1}^{S_k}, \tag{18}$$

where $S_k$ denotes the number of spikes generated by the $k$-th component, with $k = 0$ denoting the background spikes.

Next, note that each of the constituent Poisson processes intensity functions are simple, in that the normalized intensities correspond to simple densities. In the case of the background, the normalized intensity is categorical in the neuron indices and uniform in time. The impulse responses, once normalized, are categorical in the neuron indices and conditionally Gaussian in time. We use these facts to sample the Poisson processes as described in Section A.

The final sampling procedure for PP-Seq is as follows. First sample the global parameters,

$$\lambda^\varnothing \sim \mathrm{Gamma}(\alpha_\varnothing, \beta_\varnothing) \tag{19}$$

$$\pi^\varnothing \sim \mathrm{Dirichlet}(\gamma_\varnothing \cdot \mathbf{1}_N) \tag{20}$$

$$\pi \sim \mathrm{Dirichlet}(\gamma \cdot \mathbf{1}_R) \tag{21}$$

$$\mathbf{a}_r \sim \mathrm{Dirichlet}(\varphi \cdot \mathbf{1}_N) \qquad \text{for } r \in \{1, \ldots, R\} \tag{22}$$

$$c_{nr} \sim \text{Inv-}\chi^2(\nu, \sigma^2) \qquad \text{for } (n, r) \in \{1, \ldots, N\} \times \{1, \ldots, R\} \tag{23}$$

$$b_{nr} \sim \mathrm{Normal}(0, c_{nr}/\kappa) \qquad \text{for } (n, r) \in \{1, \ldots, N\} \times \{1, \ldots, R\}. \tag{24}$$

Then sample the latent events,

$$K \sim \mathrm{Poisson}(\psi \cdot T) \tag{25}$$

$$r_k \sim \mathrm{Categorical}(\pi) \qquad \text{for } k \in \{1, \ldots, K\} \tag{26}$$

$$\tau_k \sim \mathrm{Uniform}(\,[0, T]\,) \qquad \text{for } k \in \{1, \ldots, K\} \tag{27}$$

$$A_k \sim \mathrm{Gamma}(\alpha, \beta) \qquad \text{for } k \in \{1, \ldots, K\}. \tag{28}$$

Next sample the background spikes,

$$S_\varnothing \sim \mathrm{Poisson}\big(\lambda^\varnothing \cdot T\big) \tag{29}$$

$$n_0^{(s)} \sim \mathrm{Categorical}(\pi^\varnothing) \qquad \text{for } s \in \{1, \ldots, S_\varnothing\} \tag{30}$$

$$t_0^{(s)} \sim \mathrm{Uniform}(\,[0, T]\,) \qquad \text{for } s \in \{1, \ldots, S_\varnothing\} \tag{31}$$

Note that we have decomposed each neuron's background firing rate as the product: $\lambda_n^\varnothing = \lambda^\varnothing \pi_n^\varnothing$. Finally, sample the induced spikes in each sequence,

$$S_k \sim \mathrm{Poisson}(A_k) \qquad \text{for } k \in \{1, \ldots, K\} \tag{32}$$

$$n_k^{(s)} \sim \mathrm{Categorical}(\mathbf{a}_{r_k}) \qquad \text{for } s \in \{1, \ldots, S_k\}; \text{ for } k \in \{1, \ldots, K\} \tag{33}$$

$$t_k^{(s)} \sim \mathrm{Normal}(\tau_k + b_{n_s r_k}, c_{n_s r_k}) \qquad \text{for } s \in \{1, \ldots, S_k\}; \text{ for } k \in \{1, \ldots, K\} \tag{34}$$

The hyperparameters of the model are $\xi = \{\gamma, \gamma_\varnothing, \alpha_\varnothing, \beta_\varnothing, \varphi, \nu, \sigma^2, \kappa, \psi, \alpha, \beta\}$, and the global parameters consist of $\Theta = \{\theta_\varnothing, \{\theta_r\}_{r=1}^R\}$, where $\theta_\varnothing = \{\lambda^\varnothing, \{\pi_n^\varnothing\}_{n=1}^N\}$ denotes parameters related to the background spiking process and $\theta_r = \{\pi_r, \mathbf{a}_r, \{b_{nr}, c_{nr}\}_{n=1}^N\}$ denotes parameters associated with each sequence type for $r \in \{1, \ldots, R\}$. The resulting dataset is the union of background and induced spikes, $\bigcup_{k=0}^K \bigcup_{s=1}^{S_k} \{(n_k^{(s)}, t_k^{(s)})\}$.

## B.2 Expected Sequence Sizes

As described in the main text, each sequence type induces a different pattern of activation across the neural population. In particular, each neuron's response is modeled by a scaled normal distribution described by three variables—the amplitude ($a_{nr}$), the mean ($b_{nr}$), and the variance ($c_{nr}$).

We can interpret $b_{nr}$ as the temporal offset or delay for neuron $n$, while $c_{nr}$ sets the width of the response. They are drawn from a *normal-inverse-chi-squared* distribution, which is a standard choice for the conjugate prior when inferring the mean and variance of a univariate normal distribution. Refer to [3] (section 3.3) and [4] (section 4.6.3.7) for appropriate background material.

We can interpret $a_{nr}$ as the expected number of spikes emitted by neuron $n$ for a sequence of type $r$ with unit amplitude (i.e., when $A_k = 1$). By sampling $\mathbf{a}_r \in \mathbb{R}_+^N$ from a symmetric Dirichlet prior, we introduce the constraint that $\sum_n a_{nr} = 1$, which avoids a degeneracy in the amplitude parameters over neurons ($a_{nr}$) and sequences ($A_k$). As a consequence, we can interpret $A_k$ as the expected number of spikes evoked by latent event $k$ over the full neural population:

$$\mathbb{E}[S_k] = \sum_{n=1}^N \int_0^T A_k \cdot a_{nr} \cdot \mathcal{N}(t \mid \tau_k + b_{nr_k}, c_{nr_k}) \, \mathrm{d}t = A_k \cdot \underbrace{\sum_{n=1}^N a_{nr}}_{=1} \cdot \underbrace{\int_0^T \mathcal{N}(t \mid \tau_k + b_{nr_k}, c_{nr_k}) \, \mathrm{d}t}_{\approx 1} \approx A_k. \quad (35)$$

This approximation ignores boundary effects, but it is justified when $0 \ll \tau_k \ll T$ and $b_{nr}, \sqrt{c_{nr}} \ll T$; i.e. when the sequences are short relative to the interval $[0, T]$. This condition will generally be satisfied.

## B.3 Log Likelihood

Given a set of latent events $\{z_k\} = \{z_k\}_{k=1}^K$ and sequence type parameters $\{\theta_r\} = \{\theta_r\}_{r=1}^R$, we can evaluate the likelihood of the dataset $\{x_s\} = \{x_s\}_{s=1}^S$ as:

$$\log p(\{x_s\} \mid \{z_k\}, \{\theta_r\}) = \underbrace{\sum_{s=1}^S \log \left[ \lambda^\varnothing \pi_{n_s}^\varnothing + \sum_{k=1}^K A_k a_{n_s r_k} \mathcal{N}(t_s \mid \tau_k + b_{n_s r_k}, c_{n_s r_k}) \right]}_{(*)} - \underbrace{\lambda^\varnothing T - \sum_{k=1}^K A_k}_{(**)} \quad (36)$$

where terms $(*)$ and $(**)$ respectively correspond to the logarithms of $\left[ \prod_i \lambda(x_i) \right]$ and $\left[ \exp[-\int_R \lambda(x) \mathrm{d}x] \right]$ appearing in Equation (8).

# C  Complete Gibbs Sampling Algorithm

In this section we describe the collapsed Gibbs sampling routine for PP-Seq in detail. It adapts "Algorithm 3" of Neal [5] and the extension by Miller et al. [6] to draw approximate samples from the model posterior. To do this, we introduce auxiliary parent variables $u_s \in \{0\} \cup \mathbb{Z}_+$ for every spike. These variables denote assignments of each spike to of the latent sequence events, $u_s = k$ for some $k \in \{0 \ldots K\}$, or to the background process, $u_s = 0$. Let $X_k = \{x_s : u_s = k\}$ denote the set of spikes assigned to sequence $k \in \{1, \ldots, K\}$ under parent assignments $\{u_s\}_{s=1}^S$. Similar assignment variables are used in standard Gibbs sampling routines for finite mixture models (see, e.g., section 24.2.4 in [4]) and for Dirichlet Process Mixture (DPM) models (see [5]; section 25.2.4 in [4]). In these contexts, each $u_s$ may be thought of as a sequence assignment or indicator variable. In the present context of the PP-Seq model, the attributions of spikes to latent events follows from the superposition principle of Poisson processes (see section A), which allows us to decompose each neuron's firing rate function into a sum of simple functions (in this case, univariate Gaussians) and thus derive efficient Gibbs sampler updates.

The Gibbs sampling routine is summarized in Algorithm 1. While the approach closely mirrors existing work by Neal [5] and Miller et al. [6] there are a couple notable differences. First, the background spikes in the PP-Seq model do not have an obvious counterpart in these prior algorithms, and this feature of the model renders the partition of datapoints only *partially* exchangeable (intuitively, the set of datapoints defined by $u_s = 0$ is treated differently from all other groups). Second, the factors $S_k + \alpha$ and $\alpha \left( \frac{\beta}{1+\beta} \right)^\alpha \psi$ in the collapsed Gibbs updates are specific to the special form of the PP-Seq model as a Neyman-Scott process; for example, in the analogous Gibbs sampling routine for DPMs,

**Algorithm 1:** Collapsed Gibbs sampling for PPSeq model

---

**Input:** Spikes $\{x_1, \ldots, x_S\}$ and hyperparameters $\xi$.

Initialize by assigning all spikes to the background: $u_s = 0$ for $s \in \{1 \ldots S\}$.

**repeat** $M$ times to draw $M$ samples

   1. Resample parent assignments, integrating over latent events:

     **for** $s = 1, \ldots, S$. Remove $x_s$ its current assignment and place it...

      a. in the background, $u_s = 0$, with probability $\propto (1 + \beta)\, \lambda_{n_s}^{\varnothing}$

      b. in cluster $u_s = k$, with probability

$$\propto (\alpha + |X_k|) \left[\sum_{r_k=1}^{R} p(r_k \mid X_k)\, a_{n_s r_k}\, p(t_s \mid X_k, r_k, n_s)\right]$$

      c. in a new cluster, $u_s = K + 1$, with probability $\propto \alpha \left(\frac{\beta}{1+\beta}\right)^{\alpha} \psi \sum_{r=1}^{R} \pi_r\, a_{n_s r}$

     **end**

   2. Resample latent events

     **for** $k = 1, \ldots, K$ sample $r_k, \tau_k, A_k \sim p(r_k, \tau_k, A_k \mid X_k, \Theta)$ with the following steps:

      a. Sample $r_k \sim p(r_k \mid X_k, \Theta, \xi)$ where

$$p(r_k = r \mid X_k, \Theta, \xi) \propto \pi_{r_k} \left(\prod_{x_s \in X_k} a_{n_s r_k}\right) \frac{Z(\sum_{x_s \in X_k} J_{sk}, \sum_{x_s \in X_k} h_{sk})}{\prod_{x_s \in X_k} Z(J_{sk}, h_{sk})}$$

      and where $J_{sk} = 1/c_{n_s r_k}$, $h_{sk} = (t_s - b_{n_s r_k})/c_{n_s r_k}$, and

$$Z(J, h) = (2\pi)^{1/2} J^{-1/2} \exp\left\{\tfrac{1}{2} h^2 J^{-1}\right\}$$

      b. Sample $\tau_k \sim p(\tau_k \mid r_k, X_k, \Theta)$ where,

$$p(\tau_k \mid r_k, X_k, \Theta) = \mathcal{N}(\tau_k \mid \mu_k, \sigma_k^2)$$

      and where $\sigma_k^2 = (\sum_{x_s \in X_k} J_{sk})^{-1}$ and $\mu_k = \sigma_k^2 (\sum_{x_s \in X_k} h_{sk})$

      c. Sample $A_k \sim \text{Ga}(\alpha + |X_k|, \beta + 1)$.

     **end**

   3. Resample global parameters

     a. Sample $\lambda_n^{\varnothing} \sim \text{Ga}(\alpha_{\varnothing} + \sum_{x_s \in X_0} \mathbb{I}[n_s = n], \beta_{\varnothing} + T)$ for $n = 1, \ldots, N$.

     b. Sample $\pi \sim \text{Dir}\left([\gamma + \sum_k \mathbb{I}[r_k = 1], \ldots, \gamma + \sum_k \mathbb{I}[r_k = R]]\right)$

     c. Sample $\mathbf{a}_r \sim$
$$\text{Dir}\left(\left[\varphi + \sum_k \sum_{x_s \in X_k} \mathbb{I}[r_k = r \wedge n_s = 1], \ldots, \varphi + \sum_k \sum_{x_s \in X_k} \mathbb{I}[r_k = r \wedge n_s = N]\right]\right)$$

     d. Sample $b_{nr}, c_{nr} \sim \text{Inv-}\chi^2(c_{nr}; \nu_{nr}, \sigma_{nr}^2)\mathcal{N}(b_{nr} \mid \mu_{nr}, c_{nr}/\kappa_{nr})$ for $n = 1 \ldots, N$, $r = 1 \ldots, R$ where

$$S_{nr} = \sum_{k=1}^{K^*} \sum_{x_s \in X_k} \mathbb{I}[r_k = r \wedge n_s = n]$$

$$\nu_{nr} = \nu + S_{nr} \qquad\qquad \sigma_{nr}^2 = \frac{\nu\sigma^2 + \sum_{k=1}^{K^*} \sum_{x_s \in X_k} (t_s - \tau_k)^2 \cdot \mathbb{I}[r_k = r \wedge n_s = n]}{\nu + S_{nr}},$$

$$\kappa_{nr} = \kappa + S_{nr} \qquad\qquad \mu_{nr} = \frac{\sum_{k=1}^{K^*} \sum_{x_s \in X_k} (t_s - \tau_k) \cdot \mathbb{I}[r_k = r \wedge n_s = n]}{\kappa + S_{nr}}.$$

**end**

---

$(\alpha + |X_k|)$ is replaced with $|X_k|$, and $\alpha \, \psi \, T \left( \frac{\beta}{1+\beta} \right)^{\alpha}$ is replaced with the concentration parameter of the Dirichlet process.

## D   Derivations

### D.1   Prior Distribution Over Partitions

First we derive the prior distribution on the partition of spikes under a Neyman-Scott process. Technically, we derive the prior distribution on spike *indices*, $\{1, \ldots, S\}$, not yet including the spike times and neurons. A partition of indices is a set of disjoint, non-empty sets whose union is $\{1, \ldots, S\}$. We represent the partition as $\mathcal{I} = \{\mathcal{I}_k\}_{k=0}^{K^*}$, where $\mathcal{I}_k \subset \{1, \ldots, S\}$ contains the indices of spikes assigned to sequence $k$, with $k = 0$ denoting the background spikes. Here, $K^*$ denotes the number of non-empty sequences in the partition, and $|\mathcal{I}_k|$ denotes the number of spikes assigned to the $k$-th sequence. Likewise, $|\mathcal{I}_0|$ denotes the number of spikes assigned to the background.

**Proposition 1.** *The prior probability of the partition, integrating over the latent event parameters, is,*

$$p(\mathcal{I} \mid \Theta, \xi) = V(K^*; \xi) \frac{\mathrm{Po}(|\mathcal{I}_0|; \lambda^{\varnothing}T) \, |\mathcal{I}_0|!}{S! \, (1+\beta)^{S-|\mathcal{I}_0|}} \prod_{k=1}^{K^*} \frac{\Gamma(|\mathcal{I}_k| + \alpha)}{\Gamma(\alpha)}, \tag{37}$$

*where $\alpha$ and $\beta$ are the hyperparameters of the gamma intensity on amplitudes, $\lambda^{\varnothing}T$ is the expected number of background events, and*

$$V(K^*; \xi) = \sum_{K=K^*}^{\infty} \mathrm{Po}(K; \psi T) \frac{K!}{(K - K^*)!} \left( \frac{\beta}{1+\beta} \right)^{\alpha K}. \tag{38}$$

*Proof.* Our derivation roughly follows that of Miller et al. [6] for mixture of finite mixture models, but we adapt it to Neyman-Scott processes. The major difference is that the distribution above is over partitions of random size. First, return to the generative process described in Section B.1 and integrate over the latent event amplitudes to obtain the marginal distribution on sequence sizes given the total number of sequences $K$. We have,

$$p(S_0, \ldots, S_K \mid K, \Theta, \xi) = \mathrm{Po}(S_0; \lambda^{\varnothing}T) \prod_{k=1}^{K} \int \mathrm{Po}(S_k; \gamma_k) \, \mathrm{Ga}(A_k; \alpha, \beta) \mathrm{d}A_k \tag{39}$$

$$= \mathrm{Po}(S_0; \lambda^{\varnothing}T) \prod_{k=1}^{K} \mathrm{NB}(S_k; \alpha, (1+\beta)^{-1}) \tag{40}$$

$$= \frac{1}{S_0!} (\lambda^{\varnothing}T)^{N_0} e^{-\lambda^{\varnothing}T} \prod_{k=1}^{K} \frac{\Gamma(S_k + \alpha)}{N_k! \, \Gamma(\alpha)} \left( \frac{\beta}{1+\beta} \right)^{\alpha} \left( \frac{1}{1+\beta} \right)^{S_k} \tag{41}$$

$$= \frac{1}{S_0!} (\lambda^{\varnothing}T)^{S_0} e^{-\lambda^{\varnothing}T} \left( \frac{\beta}{1+\beta} \right)^{K\alpha} \left( \frac{1}{1+\beta} \right)^{S-S_0} \prod_{k=1}^{K} \frac{\Gamma(S_k + \alpha)}{S_k! \, \Gamma(\alpha)}. \tag{42}$$

Let $\{u_s\}_{s=1}^{S}$ denote a set of sequence assignments. There are $\binom{S}{S_0, \ldots, S_K}$ parent assignments consistent with the sequence sizes $S_0, \ldots, S_K$, and they are all equally likely under the prior, so the conditional probability of the parent assignments is,

$$p(\{u_s\}_{s=1}^{S} \mid K, \Theta, \xi) = \frac{1}{S_0!} (\lambda^{\varnothing}T)^{S_0} e^{-\lambda^{\varnothing}T} \prod_{k=1}^{K} \frac{\Gamma(S_k + \alpha)}{S_k! \, \Gamma(\alpha)} \left( \frac{\beta}{1+\beta} \right)^{\alpha} \left( \frac{1}{1+\beta} \right)^{N_k} \binom{S}{S_0, \ldots, S_K}^{-1} \tag{43}$$

$$= \frac{1}{S!} (\lambda^{\varnothing}T)^{S_0} e^{-\lambda^{\varnothing}T} \left( \frac{\beta}{1+\beta} \right)^{K\alpha} \left( \frac{1}{1+\beta} \right)^{S-S_0} \prod_{k=1}^{K} \frac{\Gamma(S_k + \alpha)}{\Gamma(\alpha)}. \tag{44}$$

The parent assignments above induce a partition, but technically they assume a particular ordering of the latent events. Moreover, if some of the latent events fail to produce any observed events, they will not be included in the partition. In performing a change of variables from parent assignments to partitions, we need to sum over latent event assignments that produce the same partition. There are $\binom{K}{K^*}K^*! = \frac{K!}{(K - K^*)!}$ such assignments if there are $K$ latent events but only $K^*$ partitions. Thus,

$$p(\mathcal{I} \mid K, \Theta, \xi) = \frac{K!}{(K - K^*)!} \frac{1}{S!} (\lambda^{\varnothing}T)^{|\mathcal{I}_0|} e^{-\lambda^{\varnothing}T} \left( \frac{\beta}{1+\beta} \right)^{K\alpha} \left( \frac{1}{1+\beta} \right)^{S-|\mathcal{I}_0|} \prod_{k=1}^{K^*} \frac{\Gamma(|\mathcal{I}_k| + \alpha)}{\Gamma(\alpha)}. \tag{45}$$

Clearly, $K$ must be at least $K^*$ in order to produce the partition.

Finally, we sum over the number of latent events $K$ to obtain the marginal probability of the partition,

$$p(\mathcal{I} \mid \Theta, \xi) = \sum_{K=K^*}^{\infty} \text{Po}(K; \psi T)\, p(\mathcal{I} \mid K, \Theta, \xi) \tag{46}$$

$$= V(K^*; \xi) \frac{(\lambda^{\varnothing} T)^{|\mathcal{I}_0|} e^{-\lambda^{\varnothing} T}}{S!\,(1+\beta)^{S-|\mathcal{I}_0|}} \prod_{k=1}^{K^*} \frac{\Gamma(|\mathcal{I}_k| + \alpha)}{\Gamma(\alpha)}, \tag{47}$$

where

$$V(K^*; \xi) = \sum_{K=K^*}^{\infty} \text{Po}(K; \psi T) \frac{K!}{(K-K^*)!} \left( \frac{\beta}{1+\beta} \right)^{K\alpha}. \tag{48}$$

$\square$

## D.2 Marginal Likelihood of Spikes in a Partition

Now we combine the prior on partitions of spike indices with a likelihood of spikes under that partition. To match the notation in the main text and Section C, let $X = \{X_k\}_{k=0}^{K^*}$ denote a partition of the spikes where $X_k = \{x_s : s \in \mathcal{I}_k\}$ denote the set of spikes in sequence $k$. In this section we derive the marginal probability of spikes in a sequence, integrating out the sequence time and type. For the background spikes, there are no parameters to integrate and the marginal probability is simply,

$$p(X_0 \mid \Theta) = \prod_{x_s \in X_0} \text{Unif}(t_s; [0, T])\, \text{Cat}(n_s \mid [\lambda_1^{\varnothing}/\lambda^{\varnothing}, \ldots, \lambda_N^{\varnothing}/\lambda^{\varnothing}]) \tag{49}$$

$$= T^{-|X_0|} \prod_{x_s \in X_0} \frac{\lambda_{n_s}^{\varnothing}}{\lambda^{\varnothing}}. \tag{50}$$

For the spikes attributed to a latent event, however, we have to marginalize over the type and time of the event. We have,

$$p(X_k \mid \Theta) = \sum_{r_k=1}^{R} p(r_k) \int_0^T p(\tau_k) \prod_{s \in \mathcal{I}_k} p(t_s, n_s \mid r_k, \tau_k, \Theta)\, d\tau_k \tag{51}$$

$$= \sum_{r_k=1}^{R} \pi_{r_k} \int_0^T \text{Unif}(\tau_k; [0, T]) \prod_{x_s \in X_k} \text{Cat}(n_s \mid r_k, \Theta)\, \mathcal{N}(t_s \mid \tau_k + b_{n_s r_k}, c_{n_s r_k})\, d\tau_k \tag{52}$$

$$= \frac{1}{T} \sum_{r_k=1}^{R} \pi_{r_k} \int_0^T \prod_{x_s \in X_k} \frac{a_{n_s r_k}}{\sqrt{2\pi c_{n_s r_k}}} \exp\left\{ -\frac{1}{2 c_{n_s r_k}} (t_s - \tau_k - b_{n_s r_k})^2 \right\} d\tau_k. \tag{53}$$

Let $J_{sk} = 1/c_{n_s r_k}$, and $h_{sk} = (t_s - b_{n_s r_k})/c_{n_s r_k}$. Then,

$$p(X_k \mid \Theta) = \frac{1}{T} \sum_{r_k=1}^{R} \pi_{r_k} \int_0^T \prod_{x_s \in X_k} a_{n_s r_k} \sqrt{\frac{J_{sk}}{2\pi}} \exp\left\{ -\tfrac{1}{2} J_{sk} \tau_k^2 + h_{sk} \tau_k - \tfrac{1}{2} h_{sk}^2 J_{sk}^{-1} \right\} d\tau_k \tag{54}$$

$$= \frac{1}{T} \sum_{r_k=1}^{R} \pi_{r_k} \prod_{x_s \in X_k} \frac{a_{n_s r_k}}{Z(J_{sk}, h_{sk})} \int_0^T \exp\left\{ -\tfrac{1}{2} (\textstyle\sum_{x_s \in X_k} J_{sk}) \tau_k^2 + (\textstyle\sum_{x_s \in X_k} h_{sk}) \tau_k \right\} d\tau_k \tag{55}$$

$$\approx \frac{1}{T} \sum_{r_k=1}^{R} \pi_{r_k} \left( \prod_{x_s \in X_k} a_{n_s r_k} \right) \frac{Z(\sum_{x_s \in X_k} J_{sk}, \sum_{x_s \in X_k} h_{sk})}{\prod_{x_s \in X_k} Z(J_{sk}, h_{sk})}, \tag{56}$$

where

$$Z(J, h) = (2\pi)^{1/2} J^{-1/2} \exp\left\{ \tfrac{1}{2} h^2 J^{-1} \right\} \tag{57}$$

is the normalizing constant of a Gaussian density in information form with precision $J$ and linear coefficient $h$. For a sequence with a single spike, this reduces to,

$$p(x_s \mid \Theta) \approx \frac{1}{T} \sum_{r_k=1}^{R} \pi_{r_k} a_{n_s r_k}. \tag{58}$$

The approximations above are due to the truncated integral over $[0, T]$ rather than the whole real line. In practice, this truncation is negligible for sequences far from the boundaries of the interval.

## D.3 Collapsed Gibbs Sampling the Partitions

The conditional distribution on partitions given data and model parameters is given by,

$$p(\mathcal{I} \mid \{x_s\}_{s=1}^S, \Theta, \xi) \propto p(\mathcal{I} \mid \Theta, \xi)\, p(\{x_s\}_{s=1}^S \mid \mathcal{I}, \Theta) \tag{59}$$

$$= p(\mathcal{I} \mid \Theta, \xi)\, p(X_0 \mid \Theta) \prod_{k=1}^{K^*} p(X_k \mid \Theta). \tag{60}$$

Let $X \setminus x_s$ denote the partition with spike $x_s$ removed. Though it is slightly inconsistent with the proof in Section D.1, let $u_s$ denote the assignment of spike $x_s$, as in the main text. Moreover, let $S_k$ denote the size of the $k$-th sequence in the partition, *not including spike $x_s$*. We now consider the relative probability of the full partition $X$ when $x_s$ is added to each one of the possible parts: the background, an existing sequence, or a new sequence.

For the background assignment,

$$p(u_s = 0 \mid x_s, X \setminus x_s, \Theta, \xi) \propto \frac{p(u_s = 0, x_s, X \setminus x_s, \Theta, \xi)}{p(X \setminus x_s, \Theta, \xi)} \tag{61}$$

$$= \frac{V(K^*;\xi)\, \frac{(\lambda^\varnothing T)^{S_0+1} e^{-\lambda^\varnothing T}}{S!\,(1+\beta)^{S-S_0-1}}\, p(X_0 \cup x_s \mid \Theta) \prod_{k=1}^{K^*} \frac{\Gamma(S_k+\alpha)}{\Gamma(\alpha)} p(X_k \mid \Theta)}{V(K^*;\xi)\, \frac{(\lambda^\varnothing T)^{S_0} e^{-\lambda^\varnothing T}}{S!\,(1+\beta)^{S-S_0}}\, p(X_0 \mid \Theta) \prod_{k=1}^{K^*} \frac{\Gamma(S_k+\alpha)}{\Gamma(\alpha)} p(X_k \mid \Theta)} \tag{62}$$

$$= \lambda^\varnothing T (1+\beta) \frac{1}{T} \frac{\lambda^\varnothing_{n_s}}{\lambda^\varnothing} \tag{63}$$

$$= (1+\beta)\lambda^\varnothing_{n_s}. \tag{64}$$

By a similar process, we arrive at the conditional probabilities of adding a spike to an existing sequence,

$$p(u_s = k \mid x_s, X \setminus x_s, \Theta, \xi) \propto \frac{\Gamma(S_k + \alpha + 1)}{\Gamma(S_k + \alpha)} \frac{p(X_k \cup x_s \mid \Theta)}{p(X_k \mid \Theta)} \tag{65}$$

$$= (S_k + \alpha)\, p(x_s \mid X_k, \Theta). \tag{66}$$

The predictive likelihood can be obtained via the ratio of marginal likelihoods derived above, or by explicitly calculating the categorical posterior distribution on types $r_k$ and then the categorical and Gaussian posterior predictive distributions of $n_s$ and $t_s$, respectively, given the type and the other spikes. The latter approach is what is presented in the main text.

Finally, the conditional probability of assigning a spike to a new sequence simplifies as,

$$p(u_s = K^* + 1 \mid X \setminus x_s, \Theta, \xi) \propto \left( \frac{\Gamma(\alpha+1)}{\Gamma(\alpha)} \right) \left( \frac{V(K^*+1;\xi)}{V(K^*;\xi)} \right) p(x_s \mid \Theta) \tag{67}$$

$$= \alpha \left( \frac{\sum_{J=K^*+1}^{\infty} \frac{1}{J!} e^{-\psi T} (\psi T)^J \frac{J!}{(J-K^*-1)!} \left( \frac{\beta}{1+\beta} \right)^{J\alpha}}{\sum_{K=K^*}^{\infty} \frac{1}{K!} e^{-\psi T} (\psi T)^K \frac{K!}{(K-K^*)!} \left( \frac{\beta}{1+\beta} \right)^{K\alpha}} \right) \left( \frac{1}{T} \sum_{r_k=1}^{R} \pi_{r_k} a_{n_s r_k} \right) \tag{68}$$

$$= \alpha \left( \frac{\sum_{K=K^*}^{\infty} (\psi T)^{K+1} \frac{1}{(K-K^*)!} \left( \frac{\beta}{1+\beta} \right)^{(K+1)\alpha}}{\sum_{K=K^*}^{\infty} (\psi T)^K \frac{1}{(K-K^*)!} \left( \frac{\beta}{1+\beta} \right)^{K\alpha}} \right) \left( \frac{1}{T} \sum_{r_k=1}^{R} \pi_{r_k} a_{n_s r_k} \right) \tag{69}$$

$$= \alpha \left( \frac{\beta}{1+\beta} \right)^\alpha \psi T \left( \frac{1}{T} \sum_{r_k=1}^{R} \pi_{r_k} a_{n_s r_k} \right) \tag{70}$$

$$= \alpha \left( \frac{\beta}{1+\beta} \right)^\alpha \psi \sum_{r_k=1}^{R} \pi_{r_k} a_{n_s r_k}. \tag{71}$$

## D.4 Gibbs Sampling the Latent Event Parameters

Given the partition and global parameters, it is straightforward to sample the latent events. In fact, most of the calculations were derived in Section D.2. The conditional distribution of the event type is,

$$p(r_k \mid X_k, \Theta) \propto p(r_k) \int_0^T p(\tau_k) \prod_{x_s \in X_k} p(t_s, n_s \mid r_k, \tau_k, \Theta) \, d\tau_k \tag{72}$$

$$\approx \pi_{r_k} \left( \prod_{x_s \in X_k} a_{n_s r_k} \right) \frac{Z(\sum_{x_s \in X_k} J_{sk}, \sum_{x_s \in X_k} h_{sk})}{\prod_{x_s \in X_k} Z(J_{sk}, h_{sk})}, \tag{73}$$

where, again, the approximation comes from truncating the integral to the range $[0, T]$, and is negligible in our cases.

Given the type, the latent event time is conditionally Gaussian,

$$p(\tau_k \mid r_k, X_k, \Theta) \propto p(\tau_k) \prod_{x_s \in X_k} p(t_s \mid r_k, n_s, \tau_k, \Theta) \tag{74}$$

$$= \text{Unif}(\tau_k; [0, T]) \prod_{x_s \in X_k} \mathcal{N}(t_s \mid \tau_k + b_{n_s r_k}, c_{n_s r_k}) \tag{75}$$

$$\propto \mathcal{N}(\tau_k \mid \mu_k, \sigma_k^2) \tag{76}$$

where

$$\sigma_k^2 = \left( \sum_{x_s \in X_k} J_{sk} \right)^{-1} \tag{77}$$

$$\mu_k^2 = \sigma_k^2 \left( \sum_{x_s \in X_k} h_{sk} \right). \tag{78}$$

Finally, the amplitude is independent of the type and time, and its conditional distribution is given by,

$$p(A_k \mid X_k, \xi) \propto \text{Ga}(A_k \mid \alpha, \beta) \, \text{Po}(S_k \mid A_k) \tag{79}$$

$$\propto \text{Ga}(A_k \mid \alpha + S_k, \beta_\varnothing + 1). \tag{80}$$

## D.5 Gibbs Sampling the Global Parameters

Finally, the Gibbs updates of the global parameters are simple due to their conjugate priors.

**Background rates**  The conditional distribution of the background rates is,

$$p(\lambda_n^\varnothing \mid X_0, \xi) \propto \text{Ga}(\lambda_n^\varnothing \mid \alpha_\varnothing, \beta_\varnothing) \, \text{Po}(|\{x_s : x_s \in X_0, n_s = n\}| \mid \lambda_n^\varnothing T) \tag{81}$$

$$\propto \text{Ga}(\lambda_n^\varnothing \mid \alpha_\varnothing + \sum_{x_s \in X_0} \mathbb{I}[n_s = n], \beta_\varnothing + T). \tag{82}$$

**Sequence type probabilities**  The conditional distribution of the sequence type probability vector $\pi$ is,

$$p(\pi \mid \{(r_k, \tau_k, A_k)\}_{k=1}^{K^*}, \xi) \propto \text{Dir}(\pi \mid \gamma \mathbf{1}_R) \prod_{k=1}^{K^*} \text{Cat}(r_k \mid \pi) \tag{83}$$

$$\propto \text{Dir} \left( \pi \mid \left[ \gamma + \sum_k \mathbb{I}[r_k = 1], \dots, \gamma + \sum_k \mathbb{I}[r_k = 1] \right] \right). \tag{84}$$

**Neuron weights for each sequence type**  The conditional distribution of the neuron weight vector $\mathbf{a}_r$ is,

$$p(\mathbf{a}_r \mid X, \{(r_k, \tau_k, A_k)\}_{k=1}^{K^*}, \xi) \propto \text{Dir}(\mathbf{a}_r \mid \varphi \mathbf{1}_N) \prod_{k=1}^{K^*} \prod_{x_s \in X_k} \text{Cat}(n_s \mid \mathbf{a}_{r_k}), \tag{85}$$

$$\propto \text{Dir}(\mathbf{a}_r \mid \boldsymbol{\phi}_r) \tag{86}$$

$$\phi_{rn} = \varphi + \sum_k \sum_{x_s \in X_k} \mathbb{I}[r_k = r \wedge n_s = n]. \tag{87}$$

**Neuron widths and delays**  The conditional distribution of the neuron delays $b_{nr}$ and widths $c_{nr}$ is,

$$p(b_{nr}, c_{nr} \mid X, \{(r_k, \tau_k, A_k)\}_{k=1}^{K^*}, \xi) \propto \text{Inv-}\chi^2(c_{nr} \mid \nu, \sigma^2) \, \mathcal{N}(b_{nr} \mid 0, c_{nr}/\kappa) \prod_{k=1}^{K^*} \prod_{x_s \in X_k} (\mathcal{N}(t_s \mid \tau_k + b_{nr}, c_{nr}))^{\mathbb{I}[r_k = r \wedge n_s = n]} \tag{88}$$

$$\propto \text{Inv-}\chi^2(c_{nr} \mid \nu_{nr}, \sigma_{nr}^2) \, \mathcal{N}(b_{nr} \mid \mu_{nr}, c_{nr}/\kappa_{nr}) \tag{89}$$

where

$$S_{nr} = \sum_{k=1}^{K^*} \sum_{x_s \in X_k} \mathbb{I}[r_k = r \wedge n_s = n] \tag{90}$$

$$\nu_{nr} = \nu + S_{nr} \tag{91}$$

$$\sigma_{nr}^2 = \frac{\nu \sigma^2 + \sum_{k=1}^{K^*} \sum_{x_s \in X_k} (t_s - \tau_k)^2 \cdot \mathbb{I}[r_k = r \wedge n_s = n]}{\nu + S_{nr}} \tag{92}$$

$$\kappa_{nr} = \kappa + S_{nr} \tag{93}$$

$$\mu_{nr} = \frac{\sum_{k=1}^{K^*} \sum_{x_s \in X_k} (t_s - \tau_k) \cdot \mathbb{I}[r_k = r \wedge n_s = n]}{\kappa + S_{nr}}. \tag{94}$$

### D.6  Split-Merge Sampling Moves

As discussed in the main text, when sequences containing of a small number of spikes are unlikely under the prior, Gibbs sampling can be slow to mix over the posterior of spike partitions since the probability of forming new sequences (i.e. singleton clusters) is very low. In addition to the annealed sampling approach outlined in the main text, we adapted the split-merge sampling method proposed by Jain et al. [7] for Dirichlet process mixture models to PP-Seq.[4] For simplicity, we implemented randomized split-merge proposals—i.e. without the additional restricted Gibbs sweeps described proposed in Jain et al. [7]. In practice, we found that these random proposals were sufficient for the sampler to accept both split and merge moves at a satisfactorily high rate.

Briefly, our method starts by randomly choosing two spikes $(t_i, n_i)$ and $(t_j, n_j)$ that are not assigned to the background partition and satisfy $|t_i - t_j| < W$ for some user-specified time window $W$. One could set $W = T$, in which case all pairs of spikes not assigned to the background are considerd; however, we will see that this will generally result in proposals that are extremely likely to be rejected, so it is advisable to set $W$ to be an upper bound on the expected sequence length to encourage faster mixing. After identifying the pair of spikes, we propose to either merge their sequences (if $u_i \neq u_j$), or, if they belong to the same sequence ($u_i = u_j = k$) we propose to form two new sequences, each containing one of the two spikes, the remaining $|X_k| - 2$ spikes are randomly assigned with equal probability.

Given the proposed split or merge move, $\mathcal{I} \to \mathcal{I}^*$, the Metropolis-Hastings acceptance probability is:

$$\min\left[1, \frac{q(\mathcal{I} \mid \mathcal{I}^*)}{q(\mathcal{I}^* \mid \mathcal{I})} \frac{p(\mathcal{I}^* \mid \{x_s\}_{s=1}^{S}, \Theta, \xi))}{p(\mathcal{I} \mid \{x_s\}_{s=1}^{S}, \Theta, \xi)}\right] \tag{95}$$

where $q(\mathcal{I}^* \mid \mathcal{I})$ is the proposal density, and $p(\mathcal{I} \mid \{x_s\}_{s=1}^{S}, \Theta, \xi)$ is given in eqs. (59) and (60). The sampling method then directly follows Jain et al. [7]. The ratio of proposal probabilities for split moves is:

$$\frac{q(\mathcal{I} \mid \mathcal{I}^{\text{split}})}{q(\mathcal{I}^{\text{split}} \mid \mathcal{I})} = \left(\frac{1}{2}\right)^{-(|X_k| - 2)} \tag{96}$$

And the ratio for merge moves is:

$$\frac{q(\mathcal{I} \mid \mathcal{I}^{\text{merge}})}{q(\mathcal{I}^{\text{merge}} \mid \mathcal{I})} = \left(\frac{1}{2}\right)^{(|X_k| - 2)} \tag{97}$$

## E  Time-Warped Sequences

To account for sequences that unfold at different speeds, we incorporate a time-warping component into the generative model. Each sequence is endowed with a *warp value*, which is drawn from a discrete distribution,

$$\omega_k \sim \sum_{f=1}^{F} \eta_f \delta_{w_f}(\omega_k), \tag{98}$$

where $\eta_f \geq 0$ and $\sum_{f=1}^{F} \eta_f = 1$ are the probabilities of the corresponding warp values $w_f > 0$. In this paper, we set

$$w_f = w_F^{-1+2(f-1)/(F-1)} \tag{99}$$

$$\eta_f \propto \mathcal{N}\left(f \mid \tfrac{F-1}{2}, \sigma_w^2\right). \tag{100}$$

The hyperparameters include the number of warp values $F$, the maximum warp value $w_F$, and the variance of the warp probabilities $\sigma_w^2$. We use an odd number of warp values so that $w_{(F-1)/2} = 1$.

The warp value changes the distribution of spike times in the corresponding sequence by scaling the mean and standard deviation,

$$t_s^{(k)} \sim \mathcal{N}(\tau_k + \omega_k b_{n_s^{(k)} r_k}, \omega_k^2 c_{n_s^{(k)} r_k}). \tag{101}$$

Note that one could also choose to linearly scale the variance rather than the standard deviation, as in a warped random walk. Ultimately, this is a subjective modeling choice, and scaling the standard deviation leads to slightly easier Gibbs updates later on.

We can equivalently endow each sequence with a latent *warp index* $f_k \in \{1, \dots, F\}$ where $p(f_k) \propto \eta_{f_k}$ and,

$$t_s^{(k)} \sim \mathcal{N}\left(\tau_k + w_{f_k} b_{n_s^{(k)} r_k}, w_{f_k}^2 c_{n_s^{(k)} r_k}\right). \tag{102}$$

Since there are only a finite number of warp values, we can treat the warp index and the sequence type as a single discrete latent variable $(r_k, f_k) \in \{1, \dots, R\} \times \{1, \dots, F\}$. This formulation allows us to re-use all of the collapsed Gibbs updates derived above, replacing our inferences over $r_k$ with inferences of $(r_k, f_k)$. Computationally, this increases the run-time by a factor of $F$.

Given the sequence types, times, and warp values, the conditional distribution of the neuron delays $b_{nr}$ and widths $c_{nr}$ is,

$$p(b_{nr}, c_{nr} \mid X, \{(r_k, \tau_k, f_k, A_k)\}_{k=1}^{K^*}, \xi) \tag{103}$$

$$\propto \text{Inv}-\chi^2(c_{nr} \mid \nu, \sigma^2) \mathcal{N}(b_{nr} \mid 0, c_{nr}/\kappa) \prod_{k=1}^{K^*} \prod_{x_s \in X_k} \left(\mathcal{N}(t_s \mid \tau_k + w_{f_k} b_{nr}, w_{f_k}^2 c_{nr})\right)^{\mathbb{I}[r_k = r \wedge n_s = n]} \tag{104}$$

$$\propto \text{Inv}-\chi^2(c_{nr} \mid \nu_{nr}, \sigma_{nr}^2) \mathcal{N}(b_{nr} \mid \mu_{nr}, c_{nr}/\kappa_{nr}) \tag{105}$$

where

$$S_{nr} = \sum_{k=1}^{K^*} \sum_{x_s \in X_k} \mathbb{I}[r_k = r \wedge n_s = n] \tag{106}$$

$$\Delta_{sk} = \frac{t_s - \tau_k}{w_{f_k}} \tag{107}$$

$$\nu_{nr} = \nu + S_{nr} \tag{108}$$

$$\sigma_{nr}^2 = \frac{\nu\sigma^2 + \sum_{k=1}^{K^*} \sum_{x_s \in X_k} \Delta_{sk}^2 \cdot \mathbb{I}[r_k = r \wedge n_s = n]}{\nu + S_{nr}} \tag{109}$$

$$\kappa_{nr} = \kappa + S_{nr} \tag{110}$$

$$\mu_{nr} = \frac{\sum_{k=1}^{K^*} \sum_{x_s \in X_k} \Delta_{sk} \cdot \mathbb{I}[r_k = r \wedge n_s = n]}{\kappa + S_{nr}}. \tag{111}$$

If instead you choose to linearly scale the variances, as mentioned above, the conditional distribution of the delays and widths is no longer inverse $\chi^2$. However, it is straightforward to sample the delays and widths one at a time from their univariate conditional distributions.

## F    Supplemental Details on Experiments

### F.1    ROC curve comparison of convNMF and PP-Seq

Simulated datasets consisted of $N = 100$ neurons, $T = 2000$ time units, $R = 1$ sequence type, sequence event rate $\psi = 0.02$, a default average background rate of $\lambda_n^{\varnothing} = 0.03$, default settings of $\alpha = 225$ and $\beta = 7.5$. Further, we set

$\kappa_{nr} = c_{nr}$ and $c_{nr} = 0.04$ so that neuron offsets had unit standard deviation and small jitter by default. For the case of "jitter noise" we modified $c_{nr}$ but continued to set $\kappa_{nr} = c_{nr}$ to preserve the expected distribution of neuron offsets.

For each synthetic dataset, we fit convNMF models with $R = 1$ component and a time bin size of 0.2 time units. We optimized using alternating projected gradient descent with a backtracking line search. Each convNMF model produced a temporal factor $\mathbf{h} \in \mathbb{R}^B_+$ (where $B = T/0.2 = 40000$ is the total number of time bins). We encoded the ground truth sequence times in a binary vector of length $B$ and computed ROC curves by thresholing $\mathbf{h}$ over a fine grid of values over the range $[0, \max(\mathbf{h})]$. We consider each timebin, indexed by $b$, a false positive if $h_b$ exceeds the threshold, but no ground truth sequence was present in the timebin. Likewise, a timebin is a true positive when $h_b$ exceeds the threshold and a sequence event did occur in this timebin.

We repeated a similar analysis with 100 (approximate) samples from the PP-Seq posterior distribution. Specifically, we discretized the sampled sequence event times, i.e. $\{\tau_k\}_{k=1}^K$ for every sample, into the same $B$ time bins used by convNMF. For every timebin we computed the empirical probability that it contained a sampled event from the model, averaging over MCMC samples. This resulted in a discretized, nonnegative temporal factor that is directly analogous to the factor $\mathbf{h}$ produced convNMF. We compute an ROC curve by the same method outlined above.

Unfortunately, both convNMF and PP-Seq parameters are only weakly identifiable. For example, in PP-Seq, one can add a constant to all latent event times, $\tau_k \mapsto \tau_k + C$, and subtract the same constant from all neuron offsets $b_{nr} \mapsto b_{nr} - C$. This manipulation does not modify the likelihood of the data (it does, however, affect the probability of the parameters under the prior, and so we say the model is "weakly identifiable"). The result of this is that both PP-Seq and convNMF may consistently misestimate the "true" sequence times by a small constant of time bins. To discount this nuisance source of model error we repeated the above analysis on shifted copies of the temporal factor (up to 20 time bins in each direction); we then selected the optimal ROC curve for each model and computed the area under the curve to quantify accuracy.

## F.2 Hyperparameter sweep on hippocampal data

We performed MCMC inference on 2000 randomly sampled hyperparameter sets, masking out a random 7.5% of the data as a validation set on every run. We fixed the following hyperparameter values: $\gamma = 3$, $\varphi = 1$, $\mu_{nr} = 0$ for every neuron and sequence type, $\nu_{nr} = 4$ for every neuron and sequence type. We also fixed the following hyperparameters, pertaining to time warping: $F = 10$, $\sigma_w^2 = 100$. We randomized the number of sequence types $R \in \{1, 2, 3, 4\}$ uniformly. We randomized the maximum warp value $w_F$ uniformly on $(1, 1.5)$. We randomized the mean sequence amplitude, $\alpha/\beta$, log-uniformly over the interval $(10^2, 10^4)$, and set the variance of the sequence amplitude $\alpha/\beta^2$ equal to the mean. We randomized the expected total background amplitude, $\lambda^\varnothing$, log-uniformly over the interval $(10^2, 10^4)$ and set the variance of this parameter equal to its mean. We randomized the sequence rate $\psi$ log-uniformly over $(10^{-3}, 10^{-1})$. Finally, we randomized the neuron response widths $c_{nr}$ log-uniformly over $(10^{-1}, 10^1)$.

We randomly initialized all spike assignments either with convNMF or the annealed MCMC procedure outlined in the main text (we observed successful outcomes in both cases). In annealed runs, we set an initial temperature of 500 and exponentially relaxed the temperature to one over 20 stages, each containing 100 Gibbs sweeps. (Recall that this temperature parameter scales the variance of $\mathrm{Ga}(\alpha, \beta)$ while preserving the mean of this prior distribution.) After annealing was completed we performed 100 final Gibbs sweeps over the data, the final half of which were used as approximate samples from the posterior. In this final stage of sampling, we also performed 1000 randomized split-merge Metropolis Hastings moves after every Gibbs sweep.

All MCMC runs were performed using our Julia implementation of PP-Seq, run in parallel on the Sherlock cluster at Stanford University. Each job was allocated 1 CPU and 8 GB of RAM.

## Footnotes

[4]See also Jain et al. [8] for the case of non-conjugate priors.