[Reviews · NeurIPS 2020]

Review 1

Summary and Contributions: This paper concerns "motif" finding in multi electrode spike train data (that is particular temporal arrangement of spikes from an ordered set of neurons, that recur in the spike train data, that is shrouded by noise, i.e., random spikes). The proposed technique's closest relative is the convolutive nonnegative matrix factorization, which "factorizes" the time binned spike train matrix into a sum of "motif" matrices and their corresponding trigger times plus noise. The parameterized optimization in that case is not difficult to see. The current paper gives convolutive nonnegative matrix factorization a Bayesian flavor. They appeal to the Neyman-Scott process as the generative model, for which they perform posterior inference. The proposed scheme is first of all, rate based. That is, the generative model is a hierarchical process where rate based motifs come together to assign each neuron a rate function. And at the next stage, the neuron fires according to the inhomogeneous Poisson process with that rate. Second, the rates are built out of Gaussian bumps according to each motif. Posterior inference is based on a collapsed Gibbs sampler.

Strengths: The primary novelty of the paper is the observation that a particular Neyman-Scott process can play the role of generating rate based motif's in multi neuron spike data. The construction of the mcmc based Posterior inference, draws from prior ideas on mixture models, and although laborious, are not particularly novel.

Weaknesses: The biggest drawback seems to be that the model is over parametrized, particularly since it relates to the spike train data via a Poisson process. And therefore (1) there is a possibility of finding spurious motifs when there are none. Experiments determining whether there is overfit should be done. (2) By definition, a motif is a motif only when it recurs in the data (otherwise anything could vacuously be a motif). The model does not address this. (3) Number of motifs: Dirichlet versus using an AIC/BIC like criterion would help validation.

Correctness: yes

Clarity: yes

Relation to Prior Work: yes

Reproducibility: Yes

Additional Feedback:


Review 2

Summary and Contributions: This paper formulates a point-process model for neural sequences (called PP-Seq), which allows a sparse and interpretable representation of neural sequences as clusters of spikes with latent sources. The model is extended to include a "time-warping factor" which is a local scaling of time axis within each sequence; this addition of extra parameters is feasible due to the sparsity of the model. The paper then develops a sampling-based inference method that is suitable for the model, making use of an annealing procedure for further efficiency. The method is tested on two synthetic datasets and two real datasets, first for basic demonstrations of method and then for more challenging applications with less stereotyped sequences (with more pronounced time-warping effect).

Strengths: This is an excellent paper that touches all the important aspects of method development. The model is well-motivated and theoretically sound; a suitable and efficient inference method is proposed; the method is tested using synthetic data with various statistics of interest; finally, the method is applied to real datasets to demonstrate the usefulness, while being compared to a benchmark. The proposed model (PP-Seq) is motivated as a continuous-time generalization of another well-grounded model for sequence detection, convolutive NMF, but it offers many novel properties that make this new model a valuable contribution. The model provides an intuitive representation of neural sequences as clusters of spikes around a latent source, and its sparse nature enables the addition of time-warping factor, a core feature of the method as demonstrated in this paper. This method will be a significant impact on the experimental neuroscience community; the problem of neural sequence detection is itself an important problem in contemporary neuroscience. Moreover, the time-warping factor in the model will allow flexible and interpretable analysis of hippocampal data, where such temporal scaling is commonly observed. The efficiency of the method is also a plus. For the computational neuroscience community, as well, the proposed point-process model will be a useful baseline for the development of other models.

Weaknesses: The choice of priors for the parameters in the model may be tricky when applying this method to a different dataset; if so, a further understanding of the role of priors in the inference would be a topic of further study. But this is a fine point and I do not really see a major weakness in this work.

Correctness: As far as I can see, the methods appear to be carefully developed and clearly described in the paper. The paper also follows a good practice of statistical modeling in general, for example selecting the hyperparameter through cross-validation, and testing the method using synthetic data.

Clarity: The paper is very well written, starting from the backgrounds and building towards the forefronts of the model, balancing technicalities and insights at the right level.

Relation to Prior Work: It is clearly discussed throughout the paper how the current work is motivated based on an existing method (convNMF), and what were the limitations of the previous method that were improved in this work (continuous-time, sparseness, additional features like time-warping). Results from the proposed method are compared to the previous method, demonstrating the differences.

Reproducibility: Yes

Additional Feedback: Great work --- it was a pleasure to read. I just have some minor comments: - L106, L112: the \Delta notation is not familiar to me; a brief explanation may help. - Eq 3: the \omega notation (I understand that this is for "warping") is confusing to someone from physics or electrical engineering background, because it feels like it should be a frequency. - Fig 3E: is the response "offset" same as the latency? ======= ** Edit ** The author response addressed my question about prior selection; it would be nice to include the discussion in the revised paper as well. I am still enthusiastic about this paper.


Review 3

Summary and Contributions: The authors present a model for finding sequences of spikes across a population of recorded neurons. The modeling framework (PP-seq) makes clever use of a Neyman-Scott process formulation with marked Poisson processes to define sequences that occur across a set of noisy neurons. The model definition allows the spikes to be partitioned into groups so that inference can be performed with a collapsed Gibbs sampler.

Strengths: The authors provide a thorough derivation of the statistical model and fitting methods and demonstrate how the model is appropriate for finding sequences in neural data. The model is shown to improve upon existing approaches, and it provides a novel take on the problem. The methodology is well-presented and the results are convincing. This method, especially with the time-warping of sequences, is likely to be very valuable to many in the neuroscience community and the work is very fitting for NeurIPS audience.

Weaknesses: One lingering question was how well does the Gibbs sampler converge? (metrics such as \hat{R} or recent improvements like split-\hat{R}). If different realizations of the chain end up giving somewhat different results (as the authors allude to, similar types of chains can get stuck), what recommendations can the authors provide for practitioners when using this method? UPDATE: The authors' response has answered some of my questions here. To better clarify my original comments, I did not expect the Gibbs sampler to converge to exactly the same result each run (and I don't think it's a catastrophic problem, just as it isn't for ConvNMF). I was more interested in a few best practices recommendations for experimentalists applying this approach (e.g., multiple runs). I remain enthusiastic of this method.

Correctness: The claims and methods presented are correct.

Clarity: The paper was very well-written. Minor: the terms in the Gibbs sampler in eq 5 were a bit hard to parse in the paper, but the steps were made clearer in the supplementary

Relation to Prior Work: The paper clearly compares to existing methodology. One minor point, line 46 mentions that convNMF uses a least-squares criterion and the abstract says this is suboptimal. I agree completely, but it would help to briefly mention here why it’s suboptimal.

Reproducibility: Yes

Additional Feedback:


Review 4

Summary and Contributions: The paper proposes a latent process model for spike trains using Neyman-Scott process. The method extends convNMF to continuous time and does fully Bayesian inference by MCMC. The paper also proposes to parallelize the MCMC to accelerate the inference. The method was tested on real neural recording of HVC and hippocampus.

Strengths: The proposed method uses continuous spiking time which addresses the shortcoming of binned spike counts in previous methods and provides a sparse representation. The proposed MCMC scheme enables fully Bayesian inference on the latent sequences.

Weaknesses: I suppose that the number of types R and number of events K are hyperparameters that require predetermined before inference. How were these numbers determined in this study? The authors claim that the proposed parallel MCMC has a decent speed. The global parameter Theta however does not benefit from the parallel execution. How was the parameters (including the time warping parameter) learned? The author feedback addressed my concerns.

Correctness: The claims, method and experiments are correct to the best of my knowledge.

Clarity: The paper is overall well written. Can you please elaborate more on how were alpha and beta adjusted slowly in the annealing procedure in Line 182?

Relation to Prior Work: It is clearly discussed how this work differs from previous contributions.

Reproducibility: Yes

Additional Feedback: Fig 1A. Can you please explain how the 2D latent events were sampled from a homogeneous Poisson process?

[Author Response · NeurIPS 2020]

We are very grateful for the thoughtful feedback provided by the reviewers. We will incorporate all minor comments, including the suggestions on notation given by reviewer 2 and the comments by reviewer 3 regarding least-squares sub-optimality and clarity regarding the Gibbs sampler. We respond the main substantive comments below.

..................................................................................................................................................................

**R1** *"The biggest drawback seems to be that the model is over parametrized... and therefore there is a possibility of finding spurious motifs... Experiments determining whether there is overfit should be done."*

Overfitting is certainly a possibility. Our approach is to generalize the cross-validation procedure used for convNMF (Mackevicius et al., 2019), which is related to a well-established procedure for PCA ("speckled holdout"; Wold, 1978). We evaluate the predictive likelihood after each MCMC sample, thus building an approximation to the full predictive posterior (see fig. 2). This agrees with standard Bayesian cross-validation practices (see, e.g., Ch. 7 of Gelman et al.).

To avoid any potential confusion, we note that our model is not "over-parameterized" in the typical sense unless the number of motif types, $R$, is larger than the number of neurons, $N$. We do not consider this regime, though it might be of interest—it could be viewed as a point process analogue to sparse dictionary learning.

..................................................................................................................................................................

**R1** *"By definition, a motif is a motif only when it recurs in the data (otherwise anything could vacuously be a motif)."*

We agree that such vacuous motifs might be identified during the training phase, but they would not lead to above-chance performance in the test set. In practice, we do not expect this to be a concern—neuroscientists will typically use pp-Seq on very long time series and with hyperparameter settings that render such vacuous motifs extremely unlikely.

..................................................................................................................................................................

**R1** *"Using an AIC/BIC like criterion would help [choosing the number of motif types]."*

Due to the non-parametric nature of our model (the number of latent events $K$ is random), is not immediately obvious to us how to compute the correction factors in AIC and BIC. These criteria are typically applied when identifying point estimates of parameters (e.g. in maximum likelihood inference), while we build an approximation to the full posterior. Nonetheless, at a high level, AIC and BIC aim to approximate something similar to cross-validation. Computational expense is the only downside we see to cross-validation, and in practice we have not found this to be prohibitive.

..................................................................................................................................................................

**R2** *"The choice of priors... may be tricky when applying this method to a different dataset"*

We expect practitioners to choose priors that reflect the appropriate order-of-magnitude of expected sequences—e.g. for the rat dataset the prior for sequence length was $\sim 10$s while for songbird data we chose $\sim 1$s. These choices were guided by our domain knowledge, which future practitioners will also draw upon. In many interesting biological datasets, the same neural sequences reoccur many times, suggesting that a suitably weak prior will be dominated by the likelihood term, and so the model should produce the desired result. Using synthetic data, we have observed that a weak, but misspecified, prior recovers the ground truth sequences—we will look for a way to incorporate this into our revised paper. Note that practitioners could also use cross-validation to compare amongst different hyperparameter choices.

..................................................................................................................................................................

**R3** *"One lingering question was how well does the Gibbs sampler converge?... If different realizations of the chain end up giving somewhat different results... what recommendations can the authors provide..?"*

In all of our experiments, the sampler converged to very similar parameter ranges. Fig 3B shows, e.g., that the number of sequence events, $K$, is very similar across three independent MCMC runs. Similar results hold for other parameters; we will add additional details in our revision. This concern is not unique to our model—slow mixing across separated modes is a well-known problem for MCMC inference with no easy solution. ConvNMF can also converge to different solutions across optimization runs. In practice, neural sequences are often salient enough to overcome these worst-case conclusions. We will also look into additional MCMC convergence diagnostics (e.g. Gelman-Rubin) for our revision.

..................................................................................................................................................................

**R4** *"I suppose that the number of types R and number of events K are hyperparameters that require predetermined before inference. How were these numbers determined in this study?"*

The number of sequence types $R$ must be specified, along with a few other hyperparameters (we propose to use cross-validation to guide these choices, as discussed above). The number of events, $K$, *does not need to be specified*—since this changes over MCMC samples we obtain an approximation to the posterior—i.e., $p(K \mid \texttt{data})$ informally.

..................................................................................................................................................................

**R4** *"The global parameter Theta however does not benefit from the parallel execution. How was the parameters (including the time warping parameter) learned?"*

Our current implementation does not use parallel computation in the global update since this step is very fast and not the primary bottleneck (in principle, some of these computations could be further parallelized). The Gibbs update over a grid of $W$ warping values is analogous to re-sampling the motif type over $R$ possibilities. Due to space constraints, the details are in Supplement E, but we will make an effort to provide more intuition in the main text in our revision.

[Meta-Review · NeurIPS 2020]

Inferring sequential structure in multivariate sparse neural signals is a very important problem for which the field needs more tools. This paper provides a principled sampling based approach with many clever tricks to tackling this significant problem efficiently and effectively.